# Ictogenesis proceeds through discrete phases in hippocampal CA1 seizures in mice

John-Sebastian Mueller [1,3], Fabio C. Tescarollo[1,3], Trong Huynh[1,2], Daniel A. Brenner [1], Daniel J. Valdivia [1], Kanyin Olagbegi[1], Sahana Sangappa [1], Spencer C. Chen[1,4] & Hai Sun [1,4] ✉

Epilepsy is characterized by spontaneous non-provoked seizures, yet the mechanisms that trigger a seizure and allow its evolution remain under-explored. To dissect out phases of ictogenesis, we evoked hypersynchronous activity with optogenetic stimulation. Focal optogenetic activation of putative excitatory neurons in the mouse hippocampal CA1 reliably evoked convulsive seizures in awake mice. A time-vs-time pulsogram plot characterized the evolution of the EEG pulse response from a light evoked response to induced seizure activity. Our results depict ictogenesis as a stepwise process comprised of three distinctive phases demarcated by two transition points. The induction phase undergoes the first transition to reverberant phase activity, followed by the second transition into the paroxysmal phase or a seizure. Non-seizure responses are confined to either induction or reverberant phases. The pulsogram was then constructed in seizures recorded from a murine model of temporal lobe epilepsy and it depicted a similar reverberance preceding spontaneous seizures. The discovery of these distinct phases of ictogenesis may offer means to abort a seizure before it develops.

Epilepsy is one of the most common neurological disorders, affecting approximately 1% of the population[1]. One third of patients continue to have seizures, despite taking multiple anti-seizure medications[2]. The epileptic brain is characterized by repeated transient seizures separated by longer more quiescent interictal periods. Ictogenesis describes the processes underlying transition from the interictal state to a seizure, which should be the prime target to supress seizure initiation. Yet, the neurobiological mechanisms of ictogenesis remain poorly understood. Most antiepileptic drugs act nonspecifically to reduce neuronal excitability and burst firing, or to reduce neurotransmitter release throughout the brain[3]. Therefore, they affect a large spectrum of brain activities other than seizures, resulting in unwanted side effects and reduced efficacy in preventing seizures[4]. To develop interventions targeting the transition to a seizure necessitates careful studies of ictogenesis.

The investigation of ictogenesis requires a seizure model with a reliably identified seizure onset zone to capture the cellular and network activities responsible for the transition from the interictal state to a seizure. While many animal models have been used to reproduce and analyze the mechanisms behind this transition to develop interventions[5,6], recent advancements in optogenetics have allowed for artefactual-free recording of EEG or local field potential (LFP) responses at the seizure onset zone during a seizure[7–9]. Here, we used this optogenetic approach to produce focal hypersynchronous activity, and to characterize the neuronal activity at the stimulated seizure onset zone as it underwent the process of ictogenesis.

To evaluate the transition from the preictal to ictal state, we optogenetically targeted the hippocampal area CA1 in the mouse in vivo as a model for focal seizure generation. We focused on CA1 excitatory neurons for four reasons: 1) the hippocampus is commonly the network substrate for temporal lobe epilepsy with focal to bilateral tonic-clonic seizures being the most severe form of seizures[10], 2) CA1 is the convergent point of the hippocampal perforant and

[1]Department of Neurosurgery, Rutgers Robert Wood Johnson Medical School, New Brunswick, NJ 08854, USA. [2]Department of Surgery, Rutgers New Jersey Medical School, Newark, NJ 07103, USA. [3]These authors contributed equally: John-Sebastian Mueller, Fabio C. Tescarollo. [4]These authors jointly supervised this work: Spencer C. Chen, Hai Sun. ✉e-mail: hs925@rwjms.rutgers.edu

temporoammonic neuronal signalling pathways[11,12], 3) the CA1 subfield is more commonly injured in human epilepsies[13], and 4) conflicting evidence exists for a focally defined ensemble of overly excited cells underlying ictogenesis[14–16].

Here, we show different excitability states of brain networks when probed by optogenetically inducing seizures, characterized by some stimulations leading to seizures while others did not. Most interestingly, EEG patterns are conserved on both micro- and macro-timescales (millisecond and second, respectively) among non-seizure responses to stimulation and seizure responses. We use computational techniques and build on recent discoveries[7,17–25], and present a method of analyzing the EEG recordings for investigating local electrographic changes leading to a seizure. Our results identify stepwise and discrete phases of neuronal activity underlying ictogenesis that are demarcated by key transition points. We further implement the pulsogram in spontaneous seizures recorded from a murine model of temporal lobe epilepsy. We provide evidence of distinguishable phases and transitions of EEG activity preceding spontaneous seizures similar to those identified among optogenetically-induced seizures.

## Results

### Focal activation of area CA1 induced seizures

We transduced putative excitatory neurons in hippocampal area CA1 in 8 mice with ChR2, and then implanted them at the injection site with an optrode for simultaneous optogenetic stimulation and EEG recording (Fig. 1a and Supplementary Fig. 1). As previously demonstrated in the hippocampus[7,17–20], optogenetic stimulation of CA1 excitatory neurons in our model reliably evoked focal hypersynchronous activity that became seizures. Typically, an electrographic seizure was induced during the first epoch of the first day that a mouse was stimulated with our optogenetic protocol. More severe seizures were induced during subsequent stimulation epochs, either on the first day or during future recording sessions. Convulsive seizures continued to be induced at a high probability up to 154 days after the injection of the adeno-associated virus (AAV) delivering ChR2 (Supplementary Table 1).

### Seizure rates were frequency dependent

To induce seizures, our stimulus paradigm consisted of 5 ms light pulses delivered at fixed frequencies (5, 10, 20 Hz; Supplementary Fig. 2) for 30 s. Each recording comprised 15 epochs at the fixed frequency (Fig. 1b, c). Altogether, we collected 158 recordings, totaling 2370 stimulation epochs that included 375 seizure responses (Supplementary Table 1). Seizures were identified by visual inspection for paroxysmal ictal discharges in the EEG and were classified based on behavioral characteristics observed in the recordings (detailed later).

Optogenetically-induced seizure efficacy was dependent on stimulation frequency (Fig. 1d–g)[7]. While 10 Hz stimulation trains induced seizures in 80% of recordings, 20 Hz stimulation trains induced seizures in 100% of recordings. Conversely, 5 Hz recordings registered the least seizures with all from one mouse, so were excluded from further analysis (Supplementary Table 1). This lower stimulation frequency limit for optogenetically-induced seizures in vivo was consistent with previously published results[7,17,18]. Overall, 10 Hz stimulation induced on average 2.0 seizures per recording, which increased to 6.8 seizures per recording with 20 Hz stimulation.

### First vs Breakthrough Seizures: Stimulation can overcome post-ictal depression to induce seizures

We found that CA1 is most susceptible to seizures when the brain has not been recently exposed to recent seizures/stimulation and the effects of post-ictal depression (Fig. 1d–g). Additionally, the first seizure induced in each recording always occurred during the first epoch of stimulation (10 Hz – 96/96; 20 Hz – 19/19). Conversely, if the first stimulation did not result in a seizure, no seizures emerged from subsequent stimulations. Hereafter, we refer to the first seizure of the day as first seizures.

We observed that it took less stimulation time to induce a first seizure compared to seizures in subsequent epochs. Quantifying the timing of seizure onset using canonical methods[26] was hindered by optogenetically-evoked responses obscuring visualization of seizure onsets during stimulation periods. Instead, time-frequency analysis (spectrogram) distinguished the stimulus-driven response from paroxysmal activity in all seizure epochs (Fig. 2 and Supplementary Fig. 3). The stimulus-driven response was characterized by horizontal lines at the fundamental and harmonics of the stimulation frequency. At a visually distinct timepoint, the harmonic lines of the optogenetically-evoked response faded and were obscured by aberrant activity unsynchronised to the stimulation (Fig. 2b, c). We term this large aberrant activity as paroxysmal activity, and the time where paroxysmal activity began as the paroxysmal point. This paroxysmal point became our surrogate for classically defined seizure onset.

From stimulation start to seizure onset, the time-to-paroxysmal point measures the amount of stimulation required to elicit a seizure, and consequently, measures the resiliency of the underlying brain network to seizures. Our results clearly show that it was faster to elicit a first seizure than subsequent seizures (Fig. 2f). Therefore, first seizures appeared to have increased the resistance of the brain to subsequent seizures consistent with post-ictal depression; however, that resistance could still be overcome by persistent stimulation, especially at the higher 20 Hz stimulation frequency.

In turn, we classify seizures occurring in epochs after first seizures as "breakthrough" seizures, because the presumed post-ictal depression put the brain in a more seizure-resistant state that the stimulation must overcome[27]. Breakthrough seizures were induced on seemingly random epochs (Fig. 1f, g). Two types of breakthrough seizures emerged: (1) seizures that terminated within 5 s after stimulation termed break-short seizures and (2) seizures that sustained longer than 5 s after stimulation termed break-long seizures (Fig. 2c–e and Table 1). Increasing from 10 to 20 Hz stimulation increased the incidence of both break-short and break-long seizures substantially (Fig. 1e). We further corroborate the paroxysmal point with behavior, by identifying the time when the animal first exhibited seizure-related behavior in the video recordings as the behavioral point, which would be equivalent to the clinical onset of a seizure. Our results show that the paroxysmal point is a good surrogate for seizure onset, and it generally precedes the behavior point by $2.0 \pm 0.2$ s (sem; Fig. 2h).

### Post-ictal depression and resultant seizure resistant state paradoxically associated with more severe breakthrough seizures

As shown previously, first seizures resulted in post-ictal depression that was accompanied by a neural substrate highly resistant to seizures. We investigated whether the higher seizure-resistant state was associated with less severe seizures. Video recordings were used to score behavioral seizure severity using a modified Racine Scale (RS – Supplementary Fig. 4)[28]. On the contrary, we found breakthrough seizures have a higher incidence of severe seizures (34% RS7, combined 10 and 20 Hz) compared to first seizures (11% RS7, $P < 0.0001$, Fisher's test; see also Supplementary Fig. 4a). Paradoxically, stimulation induced more severe seizures in a brain state more resistant to ictogenesis.

We examined the relationship between seizure severity and duration. Seizure duration was measured from the paroxysmal point to the (electrographic) seizure end point. A positive correlation existed between seizure severity and seizure duration for first and break-long seizures. Although break-long seizures were generally longer lasting than first seizures (Fig. 2g), the correlation coefficients between seizure severity and duration were not significantly different between the two types of seizures when accounting for RS scores ($P = 0.1118$, ANOVA, Supplementary Fig. 4b). Therefore, even though break-long

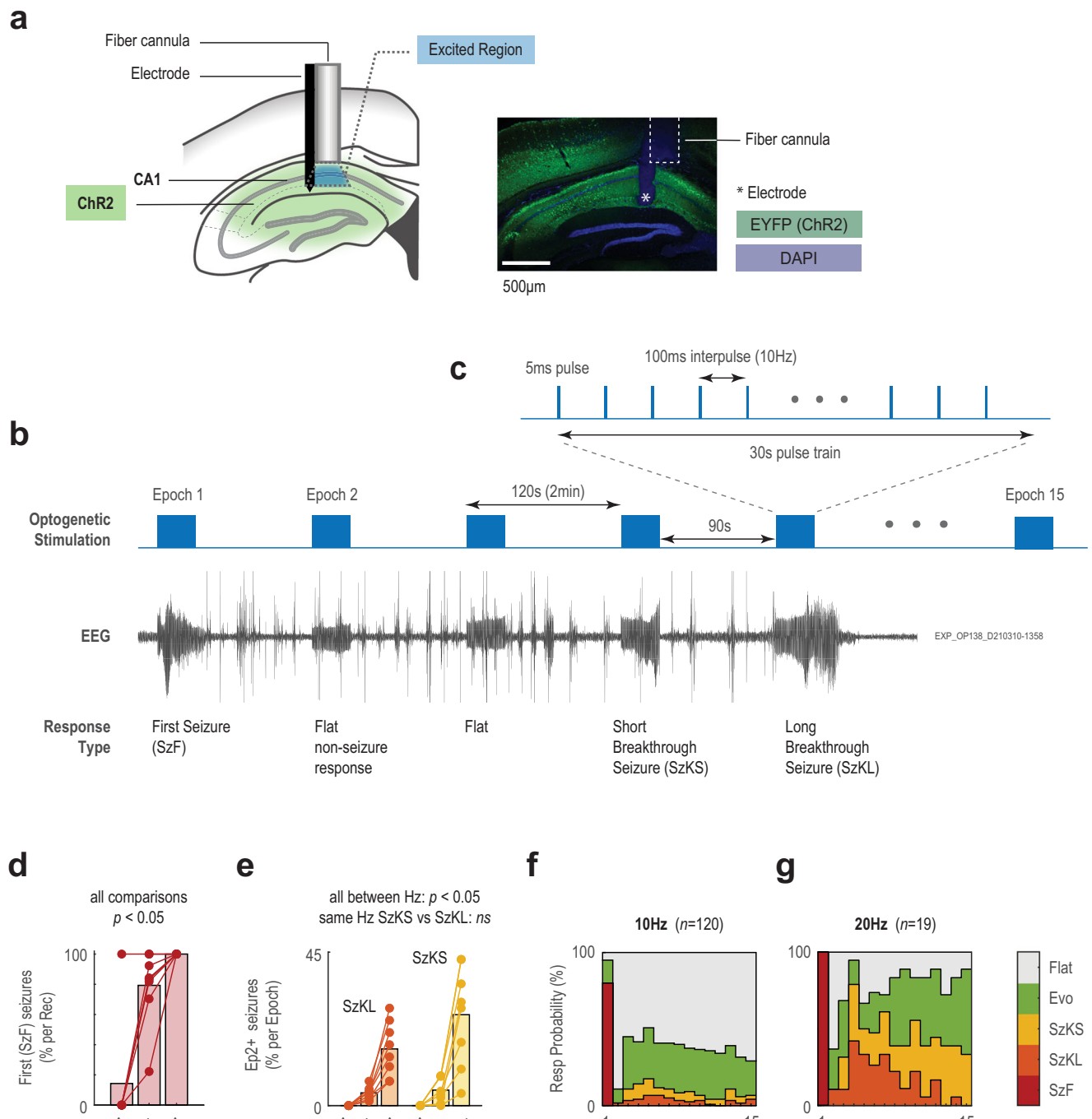

**Fig. 1 | Response types from multi-epoch optogenetic stimulation protocol delivered to mouse area CA1 putative excitatory neurons. a** Illustration of viral transduction of hippocampus, and fiber and recording electrode targeted to CA1. Dotted outline represents calculated depth of ChR2 excitation (see Methods). For comparison, a representative DAPI-stained coronal section from the hippocampus of a mouse used in experiments is included (n = 8 mice, see Supplementary Fig. 1). The dashed box indicates the optic fiber, positioned above the oriens layer of CA1 for all animals. The white asterisk indicates the tip of the implanted electrode, positioned between the radiatum and lacunosum molecular layer. **b** Diagram of 15 epoch recording paradigm with example EEG response from animal OP138. In this experiment, seizures were identified in epochs one, four and five. Flat responses to stimulation were identified in epochs two and three. **c** Schematic of a 30 s 10 Hz pulsed stimulation epoch. **d** Incidence of first seizures per recording averaged across mice (n = 8). Differences between stimulation frequency are significant (P = 0.03125 for all comparisons; two-sided Wilcoxon signed rank across mice, no adjustment for multiple comparisons). **e** Incidence of break-short (yellow) and break-long (orange) seizures per recording averaged across mice (n = 8). Differences between stimulation frequency for the same seizure types are significant (SzKL: $P_{5vs10}$ = 0.01563, $P_{5vs20}$ = 0.01563, $P_{10vs20}$ = 0.007781; SzKS: $P_{5vs10}$ = 0.03125, $P_{5vs20}$ = 0.01563, $P_{10vs20}$ = 0.007781; two-sided Wilcoxon signed rank across mice, no adjustment for multiple comparisons). **f** Distribution of response types by epoch number from 10 Hz stimulations pooled across all mice (n = 120 recordings; 96/120) color-coded by response type. **g** As in **f**, but for 20 Hz stimulations (n = 19 recordings; 19/19). Source data available at https://doi.org/10.5281/zenodo.8274424.

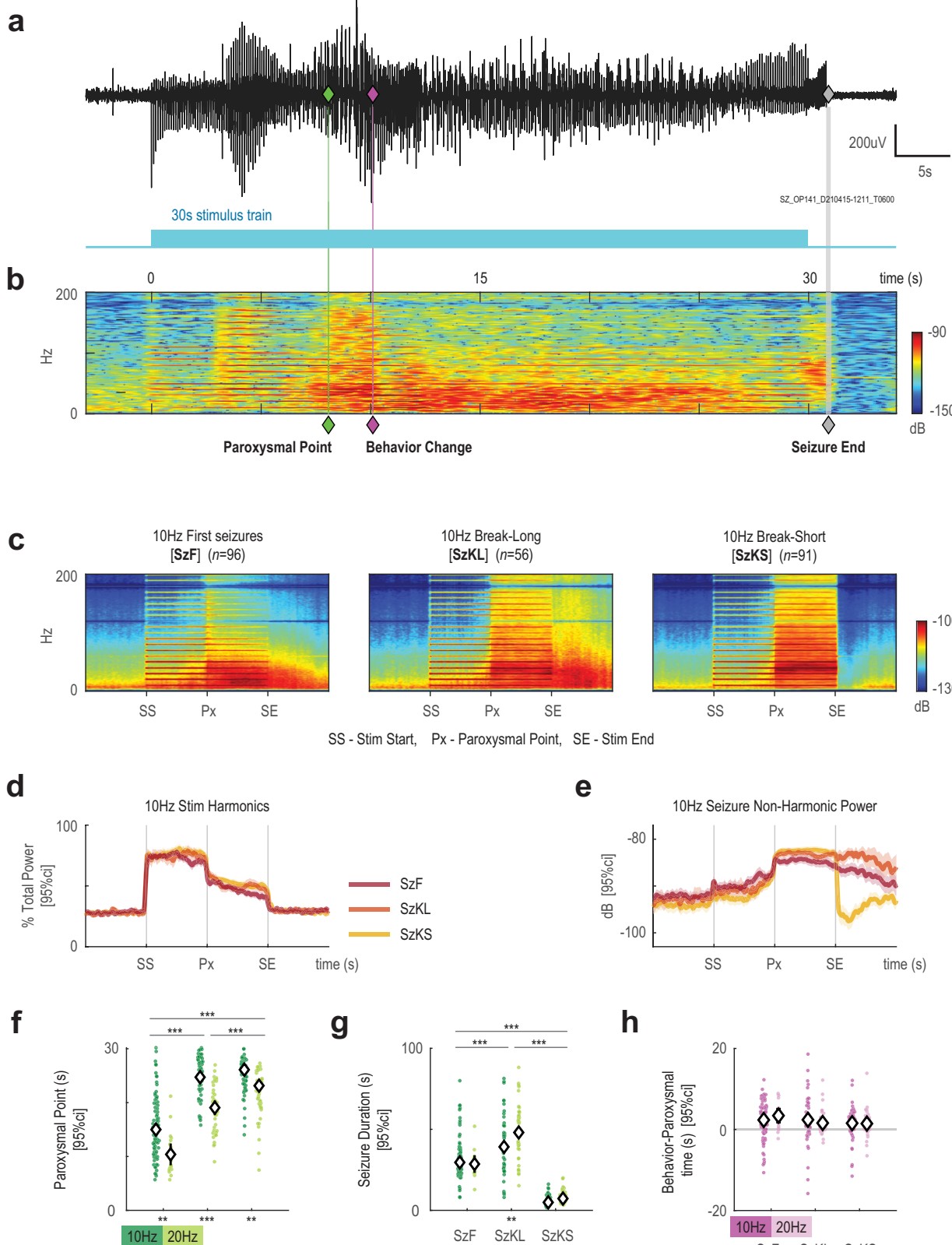

**d** 10Hz Stim Harmonics
SzF
SzKL
SzKS

**e** 10Hz Seizure Non-Harmonic Power

SS - Stim Start, Px - Paroxysmal Point, SE - Stim End

seizures were associated with a brain state more resistant to ictogenesis, the time within the paroxysmal phase necessary for severe seizures seemed similar between break-long and first seizures. Taken together, we believe that the resistance to seizures in the brain state induced by the first seizure manifested itself through changes in the pre-ictal stage of stimulation.

By contrast, there was a clear dichotomy between the severity of break-short and break-long seizures. Most break-short seizures had no behavioral manifestation (RS0 and RS1) and were brief. The difference in duration between break-long and break-short seizures suggest that seizure activity may need to be longer than 25 s in order to have behavioral manifestations of RS2 and above (Supplementary Fig. 4b).

**Fig. 2 | Spectrogram analysis of an optogenetically-induced seizure.**
**a** Representative EEG response where the green diamond demarcates the paroxysmal point, magenta diamond marks the behavioral point, and gray diamond the seizure end point. **b** Spectrogram visualization of the same EEG response in **a**. The paroxysmal point corresponds to a dramatic increase in power and the loss of the 10 Hz harmonics, and the seizure end point as a broad-spectrum decrease in power. Schematic shows the 30 s stimulation train. **c** Average spectrograms of the three identified seizure types from 10 Hz stimulation. To calculate the average, the stimulation period before and after the paroxysmal point (Px) was rescaled to an equal length for all seizures. **d** Relative power contribution of the 10 Hz harmonics over the course of the stimulation. A sharp decrease in the 10 Hz harmonics marks the paroxysmal point for all seizure types. **e** The power of the non-harmonic spectral components (0–200 Hz total power minus stimulation-driven harmonics in **d**). For Break-Short seizures, a decrease in power is observed soon after the end of the stimulation while the elevated power persists for Break-Long and First

seizures. For **d** and **e** the mean is plotted with shaded 95%CI; $n$ as labeled in **c**. **f** The delay of the paroxysmal point from epoch start grouped by seizure type and then stimulation frequency (from left to right $n = 96, 19, 56, 44, 91, 67$; $P_{Hz} = 4.036 \times 10^{-15}$, $P_{Sz} = 9.845 \times 10^{-51}$). **g** As in **f**, duration of the seizure (from paroxysmal point to seizure end point; same $n$ as in **f**; $P_{Hz} = 0.0190$, $P_{Sz} = 1.332 \times 10^{-78}$). **h** As in **f**, the timing of the behavior point relative to the paroxysmal point. Diamonds mark mean of each group (from left to right $n = 89, 19, 54, 43, 53, 47$; $P_{Hz} = 0.9210$, $P_{Sz} = 0.1371$. Electrographic seizures with no behavioral point were excluded. For **f**–**h**, statistical testing was performed with two-factor ANOVA of each variable against stimulus type and frequency, two-sided, with Tukey-Kramer correction for multiple comparisons). Full ANOVA output, F-stats, and exact $P$–values are provided in Supplementary Tables 3, 4. ANOVA estimated means and 95%CI are plotted. *$P < 0.05$, **$P < 0.01$, ***$P < 0.0001$. Source data available at https://doi.org/10.5281/zenodo.8274424.

Although evaluating kindling was not central to this study, our dataset suggests seizure severity increased with multiple recording sessions, as observed in other studies of optogenetically-induced seizures[4,20,24,25]. We observed positive correlations between numbers of recording sessions and seizure severity for both first and break-long seizures (Supplementary Fig. 4c–e), consistent with kindling. However, time courses of severity increase differed significantly between first and break-long seizures. Again paradoxically, seizures occurring after post-ictal depression (break-long) actually kindled faster than seizures occurring in a brain state more prone to seizures (first). Measured by the time to reach RS6 level, 50% of animals reached RS6 after, on average, 15 recording sessions for first seizures as opposed to 5 sessions for break-long seizures. This rapid increase in break-long seizure severity is similar in timeframe to the rapid increase in seizure severity observed with short-interval kindling[29–31].

## Discrete phases of ictogenesis

The process by which local hypersynchronous activity becomes a seizure during ictogenesis is not well understood, and the traditional electrical stimulation model has not been able to map the EEG response during this ictogenesis process due to the electrical artifacts obscuring the EEG response. We observed that optogenetically-evoked responses evolved over time and could be visualized on a pulse-locked timescale[7,18]. We further engineered this method into the "pulsogram" to analyze neuronal dynamics during seizure onsets (Fig. 3 and Fig. 4). This time-time plot simultaneously illustrated immediate time-locked electrographic response dynamics on the millisecond timescale on the y-axis (pulse timescale), evolution of these changes within the stimulation epoch over the seconds timescale on the x-axis, and relative changes in voltage as color intensity.

The pulsogram revealed the evolution of the EEG response during focal seizure ictogenesis. This is a process that occurred over several seconds, with clear stepwise transitions. Taken together with the spectrogram, we could segment the process into three general phases for all recorded seizures: induction phase, reverberant phase, and paroxysmal phase. First, the pulsogram revealed a feature observed in all epochs. Each light pulse evoked a stereotypical immediate discharge – a direct electrographic response lasting approximately 10 ms after each light pulse (Fig. 4b). This response pattern was visualized as horizontal line patterns on the pulsogram, the first phase shared by all pulsograms we generated. We termed this the induction phase of ictogenesis. Second, the pulsogram revealed the emergence of the secondary discharge that was also time-locked to the light pulse but appeared 20–50 ms after each light pulse. This appearance of the secondary discharge marked the divergent point and the start of the reverberant phase of ictogenesis. Finally, the reverberant phase transitioned into a third phase where the EEG was overtaken by paroxysmal activity unsynchronized to the stimulation pulse (Fig. 4b–d). We termed this third and final phase the paroxysmal phase. The transition

from the reverberant to paroxysmal phase is the paroxysmal point, the same point used previously to mark the start of seizures.

Since the start of the paroxysmal phase—the paroxysmal point— was previously identified as the onset of our seizures, the induction and reverberant are both pre-ictal phases of ictogenesis. As opposed to the unsynchronized ictal activity, EEG responses during these pre-ictal phases were time-locked to the stimulation pulse with either fixed (immediate discharge) or variable (secondary discharge) delays. We found that the induction phase was significantly shorter for first seizures than for both breakthrough seizures, and the reverberant phase was also shorter for first seizures than break-short seizures (trend for break-long; Fig. 4e–g). This finding supports our previous assertion that first seizures likely arose in a less resistant brain state than breakthrough seizures, and that the pre-ictal phases of ictogenesis reflected changes in this resistance.

## Secondary discharge was the hallmark of the reverberant phase

The secondary discharge was the most distinctive pre-ictal feature of our discovery. In the macro time scale, its abrupt emergence in the EEG came after several seconds of stimulation (Fig. 5a). In the micro time scale, its delayed appearance after the light pulse by 20–50 ms seemed reverberant with the immediate discharge (Fig. 4b). Therefore, we named the second phase of ictogenesis the reverberant phase. We examined the relative change in power about the divergent point, and specifically the average EEG power of the first 20 ms immediately after each light pulse and that between 20–50 ms. The divergent point was characterized by a sharp rise in EEG power during the 20–50 ms period after the light pulse, representing the emergence of the secondary discharge. Conversely, the EEG power of the immediate discharge (<20 ms) appeared stable throughout the induction and reverberant phases (Fig. 5d–h and Supplementary Fig. 5). These patterns were conserved among all three seizure types: first, break-short, and break-long. These findings further reinforced that ictogenesis observed among our optogenetically-induced seizures was not a gradual process, but rather a stepwise process marked by distinctive phases demarcated by clear transition points.

## Induction and reverberant phases are necessary but insufficient for ictogenesis

We next ask whether the point of no return for ictogenesis was captured by any of the transition points that we have identified. For this, we analyzed the non-seizure epochs in our recordings. There were two types of non-seizure epochs among all recordings from our experiments. The first type contains epochs where immediate discharges were the only response to light stimulation; and the second type contains epochs where the response to the light stimulation began with the immediate discharges followed by the appearance of the secondary discharge, but the response did not reach the paroxysmal point. We termed these two types of non-seizure responses as flat

**Table 1 | Three phases of neuronal excitability**

| Phase | Identifying Characteristic | Response Types Observed In | | | | |
|---|---|---|---|---|---|---|
| | | Flat | Evo | SzKS | SzKL | SzF |
| Core | Repeated unvarying immediate discharge time-locked 0–10 ms from optogenetically-evoked pulse onset. | ✓ | ✓ | ✓ | ✓ | ✓ |
| Reverberant | Divergent point delineates onset of secondary discharge which is time-locked 20–50 ms from optogenetically-evoked pulse onset and may evolve with continued stimulations | | ✓ | ✓ | ✓ | ✓ |
| Paroxysmal | Paroxysmal point delineates the disturbance of both the immediate and secondary discharges and onset of high amplitude activity that isn't time locked. | | | ✓ | ✓ | ✓ |

| Abbreviation | Full Name (Short Name) | Identifying Characteristic |
|---|---|---|
| Flat | Flat non-seizure response (Flat) | Core activity phase |
| Evo | Evolving non-seizure response (Evolving) | Core and reverberant activity phases |
| SzKS | Short breakthrough seizure (Break-short) | Epoch 2+ seizure with end point within 5 s of the end of stimulation that contains core, reverberant, and paroxysmal activity phases |
| SzKL | Long breakthrough seizure (Break-long) | Epoch 2+ seizure with end point beyond 5 s of the end of stimulation that contains core, reverberant, and paroxysmal activity phases |
| SzF | First seizure of the day (First) | Epoch 1 seizure that contains core, reverberant, and paroxysmal activity phases |

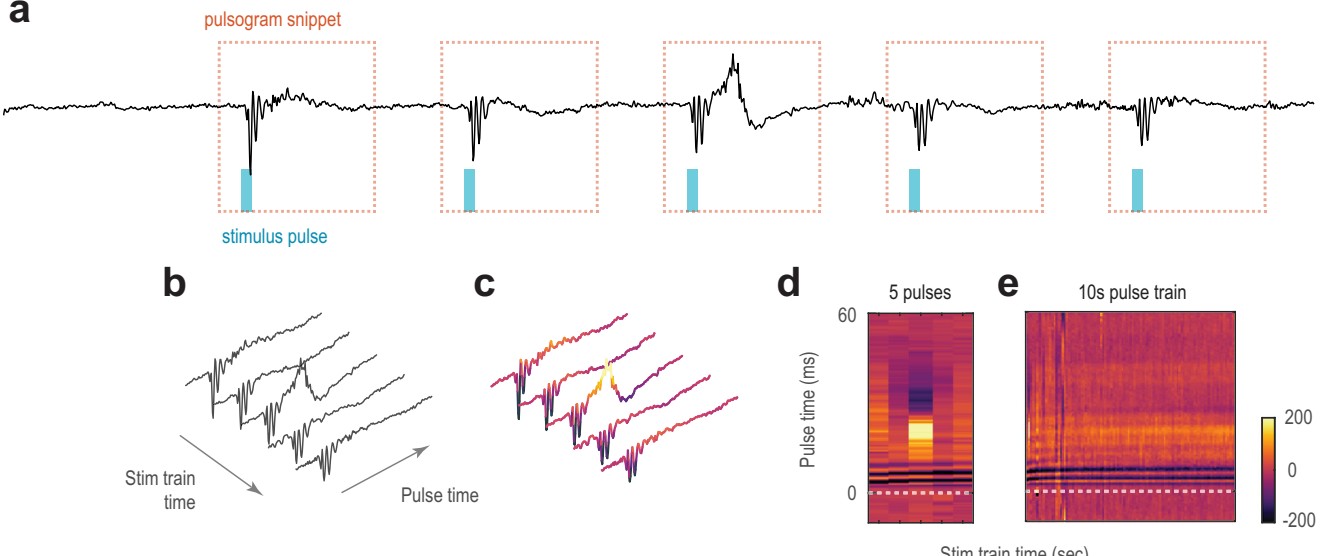

**Fig. 3 | Construction of a pulsogram. a** EEG response to optogenetic pulses, where the timing and duration of the light delivered (cyan box) and −5 to 60 ms snippet of response centered on each stimulus pulse (orange dotted box) was extracted to construct the pulsogram. In this representative EEG, stimulus frequency was 10 Hz with the standard pulse width of 5 ms. **b** Three-dimensional representation of the pulsogram, which was constructed by aligning each snippet by the time of pulse delivery in chronological order of pulse delivery. **c** Pulsogram coloring applied to pulse-aligned EEG snippets from **b**. **d** Two-dimensional view of pulse-aligned EEG snippets (pulsogram) from **c**, where the y-axis is pulse time (milliseconds), and the x-axis is duration of the stimulation train (seconds). **e** As in **d**, but the pulsogram has been extended to include 10 s of stimulation. Source data available at https://doi.org/10.5281/zenodo.8274424.

responses and evolving responses, respectively (Table 1). Together with first, break-long, and break-short seizures, all five of our observed responses have distinct electrographic patterns that could be easily differentiated qualitatively with minimal training (Fig. 5).

The similarity of activity phases between non-seizure and seizure responses from the same recording sessions was striking (Fig. 5a). We performed principal component analysis (PCA) to capture the pulsogram trajectories in their two most representative dimensions (Fig. 5b, c). It was clear that the same sequential neuronal response circuit was being traversed progressively by the flat and evolving non-seizure responses, and the short and long breakthrough seizures. These features remained largely consistent over a week and were still present when the principal components were averaged among epochs of the same response type (Supplementary Fig. 6). Over the course of experimentation, however, we observed noticeable changes in the pulsograms (Supplementary Fig. 7); this was expected from physiological adaptation to a chronic implant, resulting in modification to the electrode impedance and efficacy of stimulation. For subsequent analyses, we therefore compared between epochs from the same recording only.

Next, we ask whether the three phases of ictogenesis that we identified are necessary steps for ictogenesis. Through manual screening, the succession of epoch response phases from flat to evolving to seizures suggests that induction and reverberant phases are necessary and are activated in sequence prior to triggering the paroxysmal or ictal activity (Table 1). To demonstrate this computationally, we used an algorithm called dynamic time warping (DTW) to elucidate the successive nesting of phases between epoch response types by analyzing the trajectory representation of electrographic responses. DTW is an algorithm that ties the corresponding ends of two trajectories and non-linearly stretches (warped) all points in between so that they best align, while maintaining the sequential order

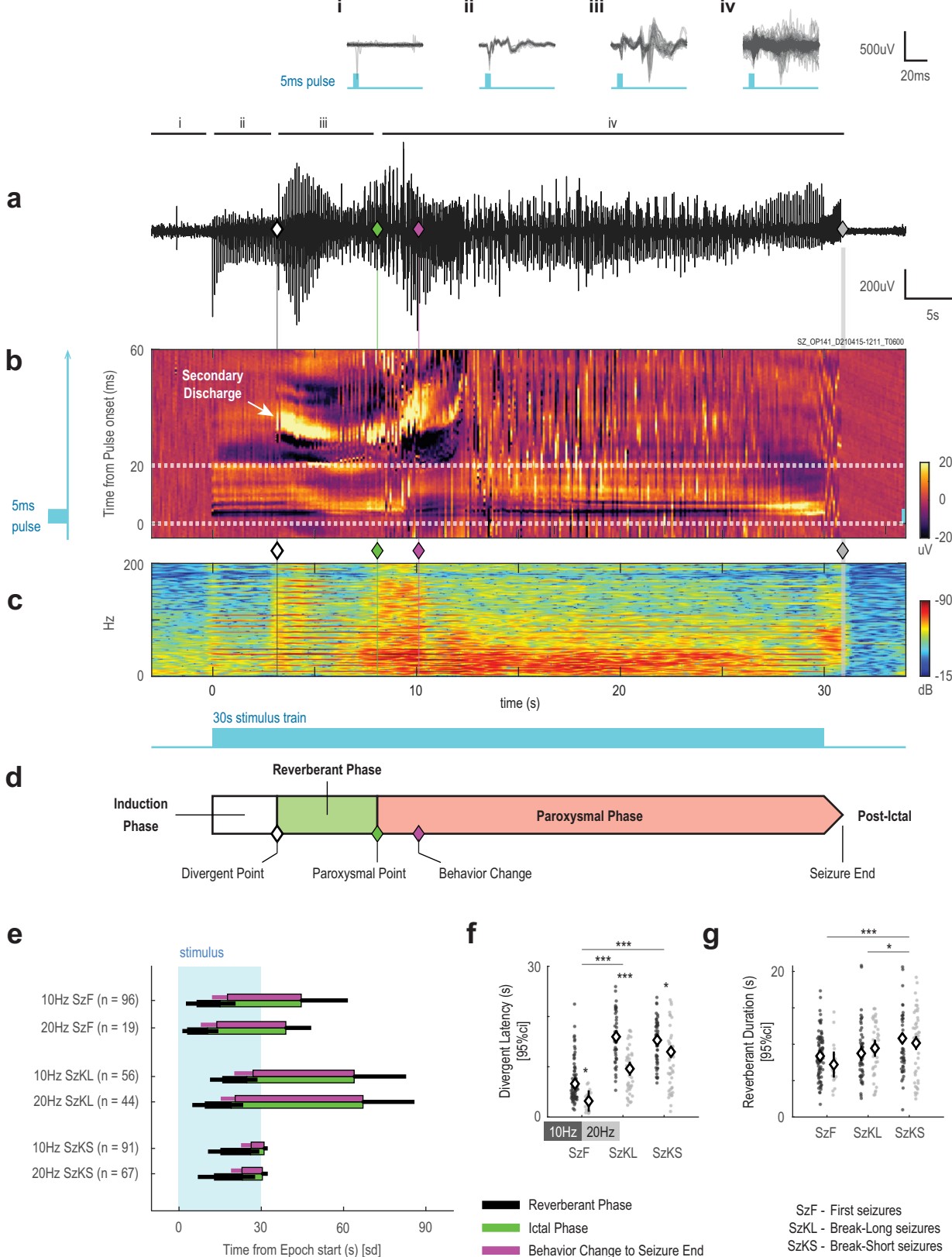

**Nature Communications** | (2023)14:6010

of the data points. The aligned trajectories were then used to calculate a distance index for the trajectory separation to measure trajectory dissimilarity. We used this warping step to examine the progressive nesting of response phases. If trajectory 1 is a subsegment of the responses contained in trajectory 2, then warping should account for most, if not all, of trajectory 1 to represent only part of trajectory 2.

By response type, we plotted the percentage of a dynamic time warped response necessary to represent another response in the same recording (Fig. 6 and Supplementary Fig. 8). The percentage of the response type being compared to all other less complex response types is plotted on the y-axis. A perfect diagonal signifies that the responses are an exact match, while a line above the diagonal means

**Fig. 4 | Phases of an optogenetically-evoked seizure. a** Full length representative EEG response from a first seizure, where the white diamond demarcates divergent point, and as in Fig. 2, green diamond for the paroxysmal point, magenta diamond marks the behavioral point, and gray diamond demarcates seizure end point. **i–iv** are traces from pre-stimulation, induction, reverberant, and paroxysmal phases, respectively. **b** Pulsogram visualization of the same EEG response in **a**. In this pulsogram, the immediate discharge occurs during the stimulation light pulse and the secondary discharge is revealed to be time-locked at an approximate 20 ms delay from the immediate discharge. (**c**) Spectrogram visualization of the same EEG response in **a**. **d** Schematic showing the 30 s stimulation train along with all 3 phases of an optogenetically-evoked seizure: induction, reverberant, and paroxysmal. **e** Mean and standard deviations of the timing from epoch start of all seizure events

and stages grouped by seizure type and then stimulation frequency. **f** The latency to the divergent point from epoch start (same $n$ as labeled in **e**; $F_{Hz}(1,367) = 50.6$, $P_{Hz} = 5.965 \times 10^{-12}$, $F_{Sz}(2,367) = 88.2$, $P_{Sz} = 5.128 \times 10^{-32}$). **g** As in **f**, duration of the reverberant phase (from divergent point to paroxysmal point; same $n$ as labeled in **e**; $F_{Hz}(1,367) = 0.59$, $P_{Hz} = 0.4419$, $F_{Sz}(2,367) = 11.04$, $P_{Sz} = 2.210 \times 10^{-5}$). For **f**, **g** statistical testing was performed with two-factor ANOVA of each variable against stimulus type and frequency, two-sided, with Tukey-Kramer correction for multiple comparisons. Full ANOVA output and exact $P$–values are provided in Supplementary Tables 3, 4. ANOVA estimated means and 95%CI are plotted. *$P < 0.05$, **$P < 0.01$, ***$P < 0.0001$. Source data available at https://doi.org/10.5281/zenodo.8274424.

the response being compared may be represented by less of the response on the y-axis. These results show that flat responses are a necessary subsegment of evolving responses, and both are subsegments of break-short seizures, which are contained within break-long seizures.

Dynamic time warping distance confirmed the stepwise dissimilarities between response types (Fig. 6a–d). Comparing the trajectory of break-long seizures, we saw that it was most similar to break-short seizures as they both contain all of induction, reverberant and paroxysmal phases. Separation distance increased when compared to evolving responses which lacked the paroxysmal phase, and increased again when compared to flat responses which further lacked the reverberant phase. These results confirmed that from the same recording, flat, evolving responses, and breakthrough seizures are progressive extensions of the same neuronal activity sequence (trajectory), and that they are successively nested such that they are sequentially and necessary progressive phases of the ictogenesis process.

### Distinct and more complex response patterns were found in first seizures

It was clear from the pulsogram (Fig. 5a) and PCA trajectories (Fig. 5b, c and Supplementary Fig. 6) that first seizures are comprised of a more complex ictogenesis pattern as captured by the reverberant phase. First seizures contained multiple loops in their PCA trajectories and had both the most complex trajectories and were most dissimilar to the other response types (Fig. 6f).

We adopted two segmentation analysis approaches to confirm the increased complexity of first seizures. First, we examined the number of distinct phases in the response with angular segments analysis, which divided the response trajectory into segments with more segments indicating increased trajectory complexity (Supplementary Fig. 9). The flat responses were the least complex, followed by the evolving response, then breakthrough, and lastly first seizures (Fig. 6e). The Ramer-Douglas-Peucker algorithm, an alternative trajectory segmentation method, substantiated these results (Supplementary Fig. 10).

The increased number of phases in first seizures could be due to the engagement of additional neural ensembles compared to breakthrough seizures, or it could be the same ensemble activated in different combinatory patterns in time. To estimate our responses' latent dimensionality, we took the PCA of each epoch's pulsogram from stimulation start to stimulation end to calculate how many PCs were required to represent 90% of the pulsogram (Fig. 6e). PCA is a known technique for dimensional reduction and broadly estimates the number of neuronal ensembles contributing to the response. Our results indicate that first seizures in general have a higher number of latent dimensions compared to breakthrough seizures, which suggests that potentially more neuronal ensembles are involved in first seizures, and their firing pattern led to the more complex phases of patterns observed in their ictogenesis process (Supplementary Table 2).

### Reverberance phase in CA1 preceded spontaneous seizures

To determine if reverberant activity preceded spontaneous seizures, we next implemented the classical murine model of temporal lobe epilepsy using intrahippocampal injection of kainic acid (Fig. 7a). Kainic acid induces status epilepticus resulting in spontaneous recurrent seizures several weeks after injection[32]. From one kainic acid animal, we identified six behavioral seizures with severity ranging from RS2 to RS7. Since we were unable to measure EEG responses time-locked to stimulation in the spontaneous behavioral seizures that we observed, a spike-detection algorithm was used to detect EEG spikes over a pre-defined threshold (Fig. 7b, c). The pulsogram was then constructed with the detected spikes. Here, the y-axis still represents the immediate time-locked electrographic dynamics on the millisecond timescale, but of the detected spikes (as opposed to an optogenetically-evoked response before). The x-axis depicts the chronological progression and evolution of detected spikes.

Similar to optogenetically induced seizures, we were able to demarcate distinguishable phases of EEG activity prior to seizure onset by examining the new pulsogram from spontaneous seizures (Fig. 7d). We identified a stable interictal spiking pattern prior to the seizure corresponding to the induction phase from optogenetic seizures. Similar to the reverberant phase in optogenetic seizures, spiking patterns in spontaneous seizures became more complex during a short period prior to entering ictal activity. The rate of spikes per second detected in the reverberant phase far exceeded the rate of spikes detected in the inter-ictal period that was analyzed 60 s before the start of the reverberant phase (Supplementary Fig. 11). In examining the power of the spontaneous seizure pulsograms, we found the transition from the interictal to the reverberant phase (divergent point) was similarly characterized by a sharp rise in EEG power during the 15–50 ms period after the start of the detected spike, representing the emergence of the secondary discharge (Fig. 7e–g). Here, the EEG power over the time range corresponding to the immediate discharge (<15 ms) appeared more stable (i.e., lower rate of increase) throughout the inter-ictal and reverberant phases. The EEG activity profiling enabled by these pulsograms provides evidence that the reverberant activity was not exclusive to optogenetically-induced seizures. Our computational approach captured a phase of neuronal network hyperactivity that is likely necessary for ictogenesis of both spontaneous and optogenetically-induced seizures.

## Discussion

We developed an in vivo optogenetic model of temporal lobe seizures to evaluate electrographic changes associated with ictogenesis and presented a method of analyzing the EEG recordings to identify discrete phases of ictogenesis in these optogenetically-induced seizures (Fig. 4), and then verified the finding in spontaneous seizures in a murine model of temporal lobe epilepsy (Fig. 7). Similar to previous studies[7,17–20], we demonstrated that sustained neuronal excitation was capable of producing a high probability of behavioral seizures in free-moving and healthy mice. Using stimulation trains repeated in short

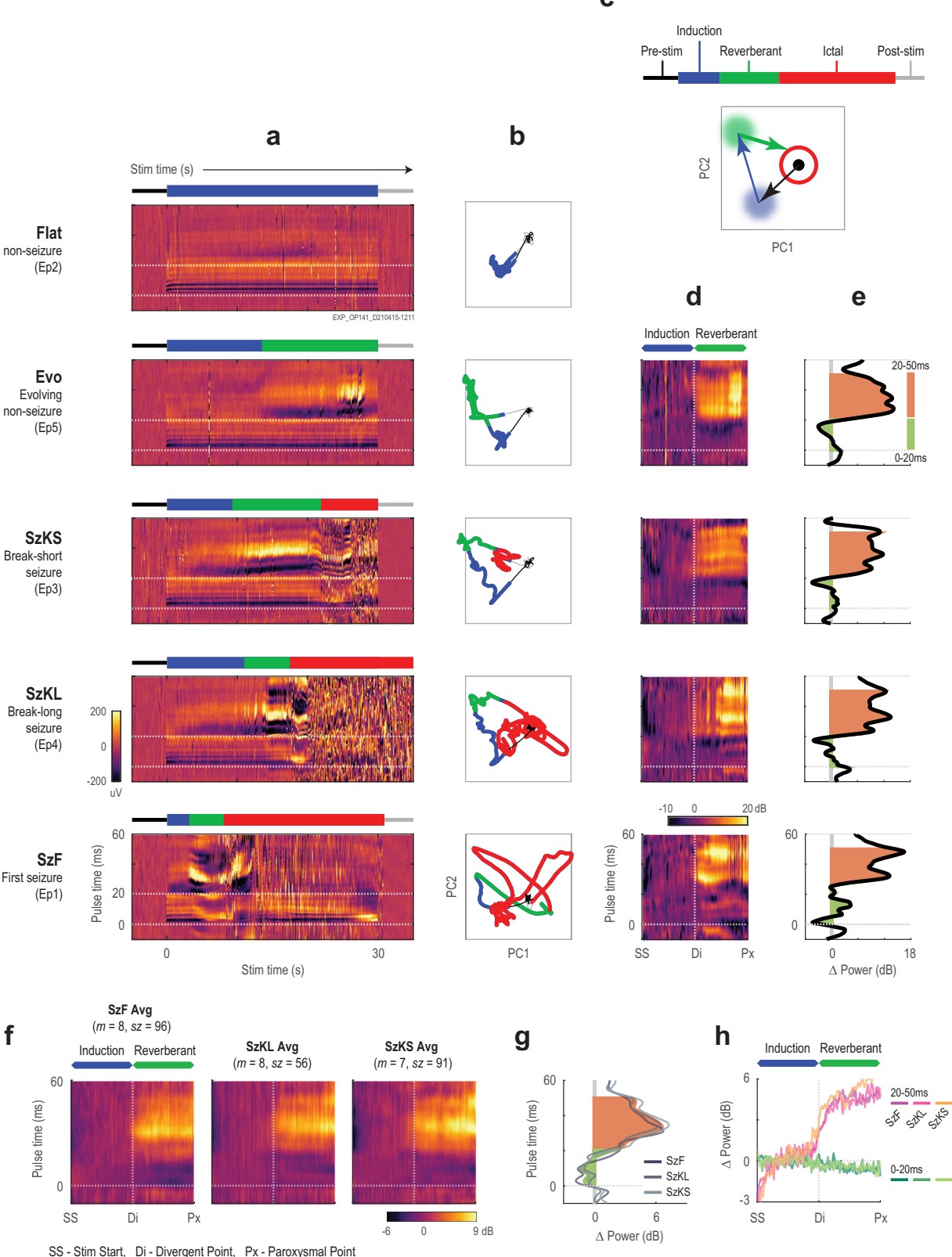

intervals, we observed two main types of seizures – first seizures from the first stimulation of each day, and breakthrough seizures in subsequent stimulations. First seizures are distinguished by more complex EEG activity sequences and progressed through the phases of ictogenesis much faster than breakthrough seizures (Fig. 2, Fig. 4, Fig. 5, and Supplementary Table 2). This first seizure, also resulted in an

increased seizure threshold, likely associated with the postictal state[27,33], or spreading depression[34]. Consequently, breakthrough seizures required longer induction (Fig. 4f) and reverberant phases (Fig. 4g) to actualize and occurred at much lower incidence. This suggests that despite these protective mechanisms, alternative routes to breakthrough to a seizure remained available, but at a higher

**Fig. 5 | Secondary discharge in seizure and non-seizure response epochs.**
**a** Representative pulsogram for EEG response from the first five epochs of an example recording from mouse OP141. Epochs were reordered to illustrate all five epoch response types in increasing order of complexity. Above each response type is a key with colors corresponding to the respective stages seen in each epoch. **b** PCA decomposition demonstrating trajectory of the pulsogram during stimulation for each epoch. **c** Schematic demonstrating the representative grouping of PC1 and PC2 components in **b** required to represent the pulsogram. **d** Power pulsogram: normalized EEG power of the smoothed pulsogram of **a**. X-axis is rescaled to align epoch start (SS), divergent (Di), and paroxysmal (Px) points for all seizure responses. EEG power was normalized by the average power during the pulse-time (horizontal line) corresponding to the induction phase, such that the average induction power is 0 dB. **e** Plot of average EEG Δpower between the reverberant

phase (divergent to paroxysmal point) and the induction phase (epoch start to divergent point), where green = area under curve from 0–20 ms after each pulse and pink = area under curve from 20–50 ms. **f** Averaged event-aligned power pulsograms (from 5**d**) for 10 Hz seizures by seizure type. Seizures from the same type were first averaged within each mouse, and then averaged across all mice ($n = 8$) for plotting. **g** Power changes for all seizures in **e** were averaged. Average Δ power for each seizure type within each mouse was first calculated and then averaged across all mice and plotted. **h** Time-dependent plot of EEG power for early pulse response (0–20 ms) compared to changes in power of the late pulse response (20–50 ms). As in **e**, the x-axis was rescaled to align with seizure events and power was normalized to the average power during the induction phase. Source data available at https://doi.org/10.5281/zenodo.8274424.

stimulation threshold. Alternate routes may partially explain why antiepileptic drugs are only effective in 60–70% of patients[35], as all routes to ictogenesis may not have been blocked by the prescribed drugs.

Breakthrough seizures increased in severity faster than first seizures. Some of the increasing severity over time may be due to the continual maturity of the ChR2 expression (up to 4–8 weeks to reach peak expression in rodents[30,31]) since we started recording at least 27 days after injection. However, the faster increase in severity of breakthrough seizures relative to first seizures should be independent of opsin expression since the evolving expression of ChR2 would similarly affect both breakthrough and first seizures. While these findings in addition to our previous findings demonstrate differences between first and breakthrough seizures (Supplementary Table 2), further investigations employing additional techniques including single-unit recordings would be necessary to determine if different cellular and network mechanisms are engaged between these two types of seizure responses. However, the same three phases of ictogenesis – induction, reverberant and paroxysmal – were found in both types of seizures. These discrete phases were discovered by using the pulsogram representation to reveal the evolution in time of the stimulus-driven and time-locked electrographic response (Fig. 4).

With the pulsograms of non-seizure responses and seizure responses from the same recording sessions juxtaposed, a clear picture of ictogenesis emerged. Neither the presence of the immediate discharge (induction phase) nor the secondary discharge (reverberant phase) is a sufficient defining feature of the point of no return of ictogenesis. Rather, these two phases identified pre-ictal sequential changes in neuronal responsivity required for ictogenesis to progress from persistent local hypersynchronous discharges to a focal seizure. The observation of non-seizure responses in our optogenetic model suggests that the neuronal tissue can be in operating states that are immune or at least highly resistant to seizures. Specifically, flat response types were resistant to secondary discharges, and evolving responses failed to advance from secondary discharges to paroxysmal activity. We demonstrated that the phases of ictogenesis, from induction to reverberant to paroxysmal, identified the necessary and sequential steps for local neuronal activity to become a seizure (Fig. 6). Such clear delineation of discrete phases of ictogenesis was not possible in the past with electrically induced seizure models[7,17,20], in which the electrical stimulation artefacts overwhelm the visualization and interpretation of the underlying neuronal response. Furthermore, the pulsogram allowed us to extrapolate more detailed electrographic characteristics from the EEG signatures of these phases of ictogenesis.

The immediate discharge, confined to the first 20 ms after each light pulse, appears to evolve in its shape over the course of the induction phase. The signal we recorded was likely a mixture of the optogenetically-evoked CA1 response and its resultant reentrant activity. Our histology review indicated that our EEG electrodes were mainly located between the stratum radiatum and lacunosum molecular layers of CA1. At this position, our recording was sensitive to

input from layer III of the entorhinal cortex via a trisynaptic reentrant loop circuit[36,37]. We cannot rule out that our stimulation activated neighboring hippocampal layers expressing ChR2. To confirm the exact composition of the immediate discharge, we would need a recording apparatus with the level of spatial resolution of in vitro patch pipettes[38], or multielectrode recording arrays[39].

One of the key findings in our analysis is the secondary discharge associated with the reverberant phase of ictogenesis, observed in both optogenetically-induced seizures and spontaneous recurrent seizures. This points to the possibility that both spontaneous and optogenetically-induced seizures may be triggered by similar underlying mechanisms that modify neuronal network excitability. Further investigations are needed to verify this hypothesis. If this is indeed true, the optogenetics based seizure model employed in our study may be used to study the ictogenesis of spontaneous seizures in experimental animals and patients with epilepsy. Here, analysis of optogenetically-induced seizures allowed for precise determination of time delays between optogenetic stimulations and the onset of secondary discharges. Based on these time delays, we propose a cellular and network mechanism behind the secondary discharge. The secondary discharge is not present during the induction phase, so unlikely results from direct light activation of stray ChR2 transductions in neurons distal to targeted regions, such as DG or CA3. Furthermore, there is no loss in amplitude of the immediate discharge at this point (Fig. 5d–h), so the secondary discharge is unlikely mediated by a response from local GABAergic networks to the abnormal neuronal activity evoked by the optical stimulation. The secondary discharge does remain time-locked to the light pulse in the reverberant phase, suggesting that it likely originates from other neurons with a reciprocal connection to those cells activated by the light. Hippocampal CA1 is a central node within the hippocampus and forms several reentrant excitatory pathways such as the previously described trisynaptic reentrant loop circuit[11,40]. Therefore, the secondary discharge could originate from another ipsilateral hippocampal loop. However, the activation of these neurons in response to the CA1 stimulation would only be delayed by a few milliseconds[41]. The 20–50 ms delay of the secondary discharge suggests that a more distant reentrant loop might be involved, such as one involving neurons in the contralateral hippocampus.

Previous optogenetics publications reported contralateral hippocampal EEG discharges at similar time frames from pulse onset[18,22]. Increasing evidence also suggests that the contralateral hippocampus is the earliest brain region recruited in mesial temporal-lobe seizures[15,17,18]. Lastly, surgical interruption of transhemispheric white matter, as in functional hemispherectomy and anterior callosotomy, have reduced generalized seizures in patients with epilepsy[42,43]. We hypothesize that the secondary discharge manifests when the immediate discharge becomes re-enforced and amplified by the contralateral hippocampus producing a resonance of approximately 15–50 Hz, a frequency range commonly observed during onsets of spontaneous seizures[44]. With sustained optogenetic stimulation, the

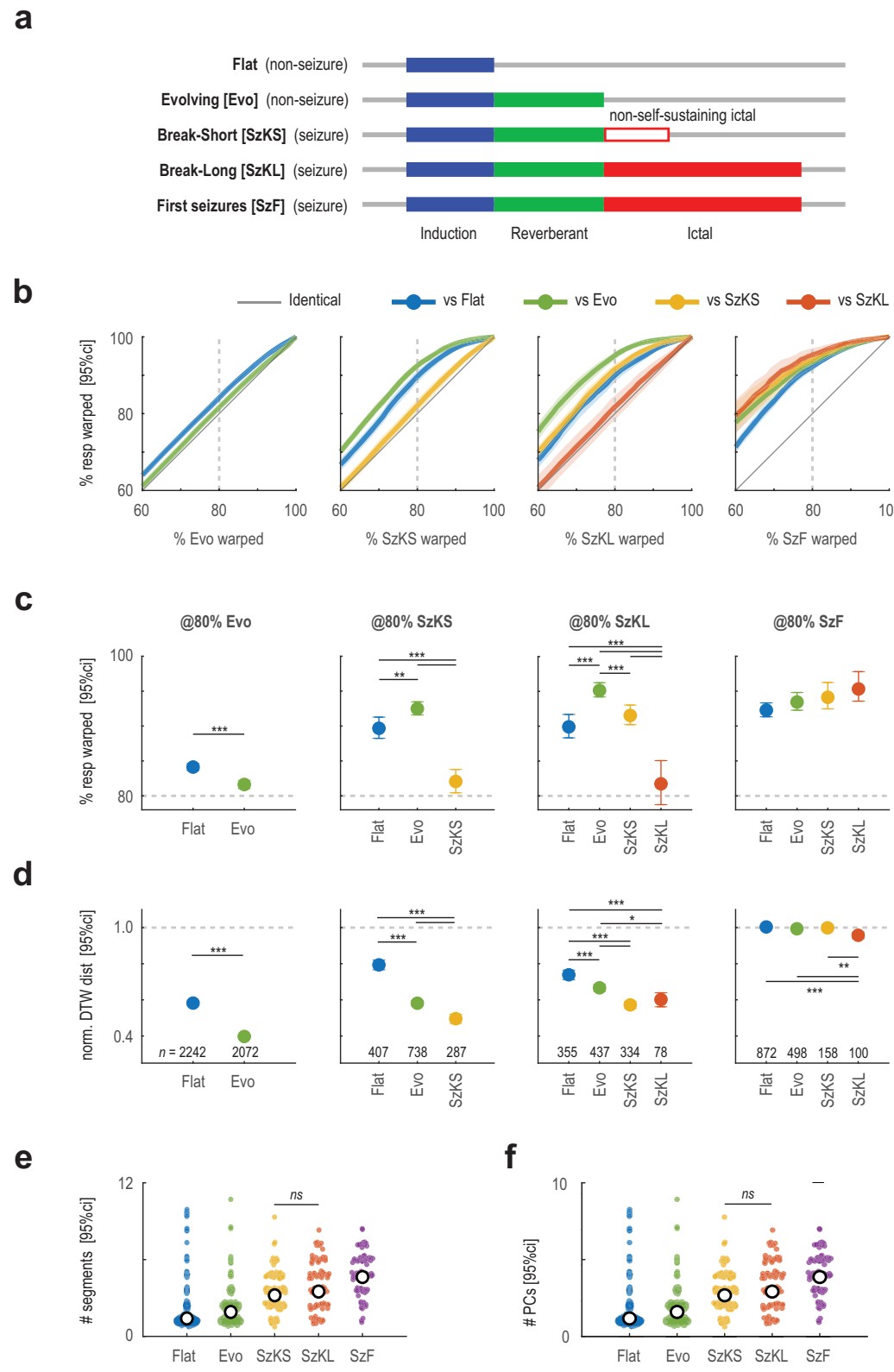

*(all comparisons are p < 0.0001 except especially marked)*

secondary discharge increased progressively in power and continued to evolve, which likely represents additional neuronal recruitment, synchrony, or an interaction between the two[45]. Future studies using inhibitory optogenetics to target the divergent point and reverberant phase would confirm the brain regions actively involved in this reentrant network[21].

As ictogenesis continues, reverberant activity built to a second transition point, beyond which the neuronal activity was no longer time-locked to optogenetic stimulation and often accompanied by a suppression of both the immediate and secondary discharges by ictal discharges (Fig. 2d, e). The transition into this paroxysmal phase designates the point where the brain no longer relies on optogenetic

**Fig. 6 | Dynamic time warping and complexity analysis of response trajectory. a** Illustration of the progressive addition of activity phases found in each of the five identified epoch responses. Note that response is aligned by activity phase rather than actual time. **b** Dynamic warping algorithm used to demonstrate embedding of activity phases from lower-order response types (*n* as labeled in **d**). The mean percentage +/− 95%CI (shaded) of the lower-order response trajectory (y-axis) required to warp into the higher-order response type (x-axis) in the same recording session. Diagonal line indicates identically matched response. Lines above the diagonal indicate a higher coverage of the lower-order response trajectory by the higher-order response (x-axis). Type of the lower-order response is color-coded. **c** % of the lower-order response required to represent 80% of the higher-order response trajectory in **b** (dashed vertical line). **d** Response separation distance calculated using dynamic time warping. Separation distance was normalized so that average distance to the first epoch response from the same recording (first seizure) was 1.0 (horizontal dashed line). **e** Result of the *n*-dimensional angular

segmentation of the response trajectories. Frequency did not have a significant effect ($F_{Hz}(1,2065) = 0.03318$, $P_{Hz} = 0.8554$). First seizures were the most complex requiring the most segments ($P < 0.0001$ against all other response types), and break-long and break-short seizures were not significantly different ($P = 0.8486$). **f** Response complexity estimated by the number of principal components (PCs) to characterize 90% of the response variance. First seizures were again the most complex ($P < 0.0001$ against all other response types). Statistics in **e, f** calculated using two-factor ANOVA against seizure type and stimulus frequency; two-sided; multiple comparison adjusted using Tukey-Kramer's method; from left to right $n = 1127, 575, 158, 100, 115$. Statistics in **c, d** calculated using Students *t*−test, two-sided, adjusted for multiple comparisons using Bonferroni's method, with *n* labeled in **d**. Statistic in **c** was calculated in z-transformed space of the % values. Statistics in **d**–**f** were calculated in the logarithm space of the values. Full ANOVA, *t*-test output and exact *P*−values are provided in Supplementary Tables 3–5. All plots show the mean +/− 95%CI. Source data available at https://doi.org/10.5281/zenodo.8274424.

stimulation to initiate bouts of abnormal neuronal activity. It corresponds to the emergence of classical after-discharges where the network entered a state where it spontaneously self-generated bursts of activity[16]. However, observation of paroxysmal activity or after-discharges did not guarantee a long duration self-sustained seizure[22], as observed in our fast-extinguishing break-short seizures (<5 s from end of stimulation). These break-short seizures may suggest that the paroxysmal activity required further progression past a third critical transition point, such as propagation into a migrating core[20,46,47], and into a fully self-sustained seizure. Though we did not identify the distinguishing feature of this third transition point in our recordings, this phenomenon presents an additional opportunity to halt ictogenesis.

Recent clinical studies suggest that seizure activity propagates from within an ictal core, or actively seizing group of cells, as penumbral regions are recruited into a seizure by an ictal wavefront[48,49]. The ictal wavefront hypothesis suggests a mechanism for seizure spread. We found similarities in our results, which investigate how the ictal core forms. In our optogenetic model, local abnormal hypersynchronous activity began in response to 10 or 20 Hz light stimulation, which we believe simulated either the origin of the ictal core after an ensemble of cells in CA1 became excited[15], or alternatively, the spread of seizure activity to involve cells in CA1 from a distant ictal core[50]. As stated earlier, it would require a recording apparatus with much higher spatial resolution for us to be able to differentiate between these two possible scenarios. Subsequently, the development and the progression of the reverberant phase may have exposed the underlying tissue to a similar process as a sweeping ictal wavefront, with a comparable time frame around 10 s. Ictal wavefronts are suspected to recruit neurons by breaking down local inhibitory restraint. In our model, a similar breakdown may explain the emergence of the secondary discharges, and its eventual development into paroxysmal activity. The formation of ictal activity at the stimulated site developed over 10–30 s in our optogenetically-induced seizure model, consistent with the "Jacksonian march" time course of seizure spread and recruitment.

It was previously suggested that single unit recording is necessary to show that neurons in the penumbra are quiescent until they have been engulfed by the ictal wavefront[46]. On the basis that a similar process occurs during the ictogenesis process that we observed, our study depicts an alternate electrophysiological approach in which EEG or LFPs may be used to identify stages of ictogenesis. LFPs are widely adopted in the surgical work up of patients with medically intractable epilepsy to locate the seizure onset zone, whereas single unit recordings are less common. Our computational techniques may be applied to the LFPs recorded from these patients to help identify preictal phases of seizure onset. This may improve our ability to accurately localize the seizure onset zone by identifying brain regions undergoing the induction and reverberant phases of ictogenesis. From our limited results, we discovered that interictal discharges seem to also undergo a stepwise

change leading into seizures (Fig. 7d). This change is characterized by an additional discharge feature that occurred with timing relative to the start of the interictal spike that was similar to that of the secondary discharge observed in optogenetically-induced seizures. Furthermore, there is a concomitant increase in the frequency of interictal discharges preceding seizures (Supplementary Fig. 11). Our ability to identify these preictal phases using LFPs alone may improve the efficacy of treating seizures using neuromodulatory techniques such as responsive neurostimulation or deep brain stimulation by allowing these devices to stimulate the brain before the seizure even begins.

## Methods

### Animals
This study was designed to analyze electrographic dynamics associated with seizure onset patterns. We used optogenetics in vivo to excite putative glutamatergic neurons in area CA1 of the hippocampus while simultaneously capturing local field potentials (LFP; used interchangeably with EEG) in CA1 in awake and freely moving mice (*n* = 8; Fig. 1a).

Animal studies were conducted per approved Rutgers Institutional Animal Care and Use Committee protocols within an American Association for Accreditation of Laboratory Animal Care accredited facility in accordance with the United States Public Health Service's Policy on Humane Care and Use of Laboratory Animals. Animals were reared and cared for under standard conditions between experiments: group-housed before surgery, single-housed with enrichments after surgery due to the implant, food and water was provided ad libitum, standardized temperature (70–74°F), humidity (30–70%RH), and housed under 12 h light/dark cycle (lights on 6am to 6 pm).

Experiments were carried out on adult male and female inbred homozygous PV-Cre mice (The Jackson Laboratory; B6.129P2-Pvalb[tm1(cre)] [Arbr]/J, RRID: IMSR_JAX:017320) with genotypes confirmed using PCR. Sex differences were analyzed post-hoc and were not found to be a significant factor for any of the results presented (Wilcoxon rank-sum); therefore, data was pooled across gender in the main analyses. PV-Cre mice were used to allow for prospective follow-up experiments utilizing Cre expression in PV interneurons. Seizure susceptibility for PV-Cre mice may be higher than wild type mice[51]; however, we do not believe optogenetic seizure induction is mechanistically different between wild type and PV-Cre mice. Animals were reared and cared for under standard conditions between experiments.

### Viral injection and optrode implantation surgery
An optrode (custom optical fiber cannula combined with recording electrode) was implanted in each animal to deliver optogenetic stimulation with simultaneous EEG recordings (Fig. 1a). The optrode included an optic fiber cannula (400μm, 0.57NA, MFC_400/430-0.66_4mm_ZF1.25(G)_FLT, Doric Lenses) housed within a 3-channel head stage connector (monopolar - model MS333/2-AIU/SPC or MS333/2-A/SPC, P1

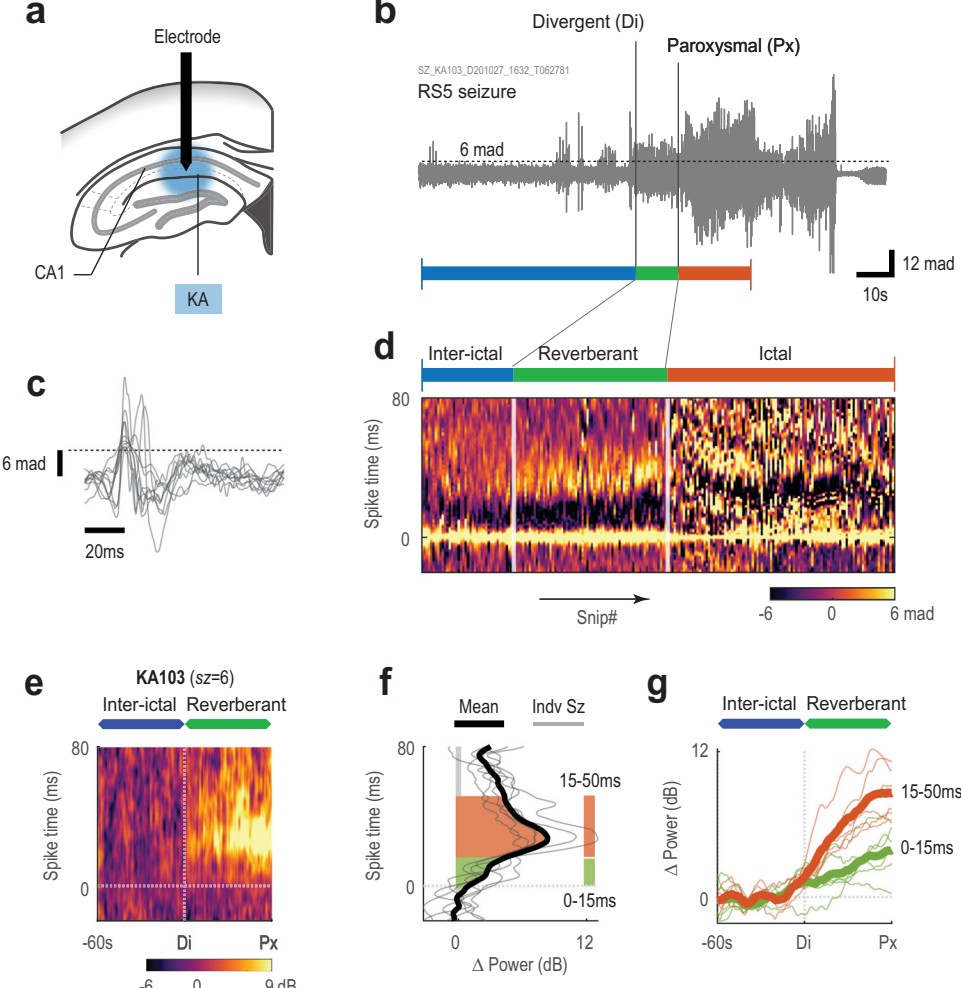

**Fig. 7 | Evidence of reverberant phase in spontaneous seizures. a** Illustration of kainic acid (KA) injection into hippocampus, and recording electrode targeted to CA1. **b** Representative EEG trace of a spontaneous seizures. Racine score (RS) label indicates the behavioral severity of the seizure (see Supplemental Fig. 4 for Racine score scale). Spontaneous seizure's equivalent of the divergent (Di) and paroxysmal (Px) points were marked from visual inspection of the EEG and constructed pulsogram in **d**. Blue line marks 60 s (inter-ictal time) leading into the divergent point. Green line marks time between divergent and paroxysmal points. Red line marks the period of the ictal phase that is shown in **d**. **c** Example EEG discharges extracted from the EEG. Discharges with peaks above 6 median-absolute-deviations (mad, dashed line) above zero were extracted. **d** Pulsogram visualization of the same EEG response in **b** by concatenating all extracted snippets such as those shown in **c**.

Note the x-axis is a non-linear progression in time by snip number. Colored lines (blue/green/red) mark the corresponding time periods in **b**. **e** Averaged event-aligned power pulsogram for spontaneous seizures ($n = 6$) as in Fig. 5f. X-axis is rescaled to align time 60 s prior to the divergent and paroxysmal points for all seizure responses. EEG power was normalized by the average power during the spike-time (horizontal line) corresponding to the inter-ictal phase, such that the average power prior to the divergent point is 0 dB. **f** Power changes for all seizures in **e** were averaged. Average Δ power for spontaneous seizures was plotted along with individual seizure Δ power. **g** Time-dependent plot of EEG power for power of the late pulse response (15–50 ms). The x-axis was rescaled to align with seizure events as in **e**. Source data available at https://doi.org/10.5281/zenodo.8274424.

Technologies or bipolar - model MS333/3-B/SPC, P1 Technologies) by a 3D-printed part. For viral injection and optrode implantation, mice were first anaesthetized with isoflurane (approximately 2.5%). Throughout the entire surgery, animals were kept warm on a heating pad while anaesthesia was maintained (approximately 1.5%). Prior to incision, bupivacaine (2.5 mg/kg) was subcutaneously injected at the incision site. Before the end of surgery, sustained release buprenorphine (EthiqaXR, 3.25 mg/kg) was injected subcutaneously to alleviate pain and discomfort. ChR2-EYFP under the CaMKIIα promoter (300 μL, pAAV5-CaMKIIα-hChR2(H134R)-EYFP, titer $2.2 \times 10^{13}$ GCmL$^{-1}$; RRID: Addgene_26969) was injected into the dorsal hippocampal CA1 layer (anteroposterior - AP: −2.06 mm, mediolateral – ML: −1.6 mm, dorsoventral – DV: −1.5 mm from bregma)[52] to selectively target putative glutamatergic neurons[53]. In the same surgery, the optrode assembly was implanted with the tip of the optic fiber 0.5 mm above, and the electrode in the viral injection site (Supplementary Fig. 1). A surface

electrode (model 51457, Stoetling) was placed over the right prefrontal cortex (AP: +1.6 mm and ML: +1.6 mm) to act as a reference. Minor variations in coordinates existed between mice; however, expression levels, fiber and electrode placement, and responses were similar for all mice so data was pooled for analysis.

Stimulation light power was measured prior to starting a recording session for each mouse (except for missing power data from OP126). Across recording sessions, light power ranged from 0.77–2.49 mW (mean and SD reported in Supplementary Table 1), measured at the tip of the fiber-optic patch cord (400μm NA0.57, Doric Lenses) using a handheld optical power meter (PM20C, Thor-Labs). At an average light power of 1.94 mW, the power at the tip of the optic cannula ferrule correlated to approximately 1.46 mW. Therefore, we estimate that we reached a depth of ~0.15 mm (based on numerical aperture of 0.57, radius of 400μm, index of refraction of brain tissue of 1.35[54], and required irradiance of 1mWmm$^{-2}$ for activating ChR2 with

blue light[55]; standalone calculation confirmed by https://web.stanford.edu/group/dlab/cgi-bin/graph/chart.php)[56]. This aligns with previously published results that targeted area CA1[57]; however, our depth is lower and half angle wider at 25° resulting in more precise and broader targeting of the pyramidal layer of area CA1.

There was variation in setting the stimulation power level between mice (Supplementary Fig. 12). In mice OP133 and OP137, we started by delivering 1.0 mW of light power. Power was gradually increased in OP137, but no clear measurable effect was found except a slight increase in response latency (Supplementary Fig. 12c). In all other mice, 2.0 mW was delivered to ensure suprathreshold stimulation. In mice OP134 and OP138, we initially maintained the same driving voltage to the LED. However, this led to a measurable degradation in output light power over time, which is likely due to minor wear on the optic fiber. We subsequently adjusted voltage levels at the beginning of each experiment to deliver 2.0 mW. Minor variations resulted from recalibration corrections. Analysis of OP137, OP138 and OP134 showed that stimulation power was not a substantial covariate in our results (Supplementary Fig. 12c); therefore, we pooled all the data for the main analyses.

## Histology
Upon experiment completion, animals were deeply anesthetized with isoflurane and then transcardially perfused with 4% paraformaldehyde. Extracted brains were dehydrated in 30% sucrose solution. 30μm thick coronal sections were prepared. To assess gross cell morphology and ChR2 expression, sections containing the dorsal hippocampus were stained using a standard DAPI (4′,6-diamidino-2-phenylindole) protocol. ChR2 expression was assessed through detection of the EYFP fluorescent tag. DAPI and EYFP fluorescence were used to confirm optrode placement (Fig. 1a, Supplementary Fig. 1).

## Optogenetic stimulations and in vivo EEG recordings
Recordings began 27 days after the surgical procedure to allow time for AAV-induced ChR2 expression (Supplementary Table 1 and Supplementary Fig. 2; except for one recording session for mouse OP126 completed on the 22nd day). ChR2 expression is expected to be stable, if not peaked, at this time point (~4 weeks post-injection)[30,31]. Animals were temporarily housed during recording sessions with tethered connections for EEG recording and optogenetic stimulus delivery. Other than the tether, the animals were awake and able to move freely. USB cameras (C920, Logitech) were used to monitor animal behavior during recording sessions. Behavior captured by synchronous video recording was used to score seizure responses (Supplementary Fig. 4).

We used a commercial off-the-shelf system (Rz10x, Tucker-Davis Technologies – TDT) to deliver optogenetic stimulation while synchronously recording EEG and video. Optogenetic stimulation was delivered with a blue, 465 nm, LED (Lx465, TDT) with pulses held at 5 ms and frequencies varied at either 5, 10, or 20 Hz (Supplementary Table 1). The total number of recordings varied between mice (see below). An individual mouse was only recorded once per day. Stimulation by different frequencies was recorded on separate days. Each recording session was comprised of 15 epochs. Each epoch consisted of a 30 s train of 5 ms stimulation pulses at the designated frequency, followed by a 90 s break between trains (Fig. 1b, c).

To investigate how lower (5 Hz) and higher (20 Hz) stimulations affect seizure induction compared to 10 Hz stimulation, a few recordings (n = 2–5 for each frequency) were completed at 5 Hz and 20 Hz upon establishing the baseline 10 Hz response (at least n = 3) and then followed by additional 10 Hz reference recordings. The days for different recordings were selected at arbitrary time points in the experimental timeline (Supplementary Fig. 2a). In four recording runs, 5 Hz stimulation preceded 20 Hz; in a separate five recording runs, the 20 Hz sessions preceded 5 Hz. We did not find any conclusive difference between the two orders of stimulation. When compared to 10 Hz recordings preceding and succeeding 20 Hz sessions, the effect of

stimulation was consistent in all mice (Supplementary Fig. 2b, c). Therefore, all analyses within the results were performed on aggregate data regardless of the order of stimulations. Note that for mice OP134, OP137 and OP138, we inserted a 10 Hz recording session after an extended (n = 5) run of 20 Hz stimulations. OP134 died from seizures during a 20 Hz stimulation session.

Simultaneous local field potential signals were obtained at 3 kHz (PZ5, TDT), band-pass filtered from 3 to 1000 Hz, and notch filtered to attenuate 60 Hz line noise and its harmonics by TDT's Rz10x. After recording sessions, EEG traces were plotted in a custom-made MATLAB application (MathWorks) and visually evaluated for optogenetically-evoked neuronal responses[18]. Noisy recordings containing movement artifacts or unstable cable connections were excluded from analysis (n = 19; Supplementary Table 1). In bipolar implanted animals, the EEG response analyzed was taken between one of the CA1 bipolar electrodes and the distant reference to maintain consistency with animals with monopolar implants. Bipolar electrodes were used in some mice to allow for future analyses (not reported here).

Spectrograms were computed for time-frequency representation of the EEG using the inbuilt function in MATLAB (0.33 s window with 87.5% overlap). To examine the spectral power around the paroxysmal point across seizures (Fig. 2c–e), the time-axis of the spectrograms for each seizure was segmented into 15 s of pre-stimulation, stimulation to paroxysmal, paroxysmal to end of stimulation, and 15 s of post-stimulation. These four segments were then rescaled to equal lengths for all seizures to take average. Harmonic power was the sum of the spectrogram at the fundamental stimulation frequency (10 Hz or 20 Hz) and its harmonics ± 1 Hz. Total power was the sum of the spectrogram across all frequencies up to 200 Hz. Non-harmonic power (Supplementary Fig. 3) is the difference between total and harmonic power.

## Intrahippocampal kainic acid injection and electrode implantation surgery
Under Isoflurane anesthesia, a 214-day-old male PV-Cre mouse (n = 1) was stereotaxically injected with kainic acid (KA – 150nL, 10 mM) unilaterally into the dorsal stratum radiatum of CA1 (AP: −2.06 mm, ML: −1.6 mm, DV: −1.6 mm from bregma). In the same surgery, a monopolar 3-channel head stage connector (model MS333/2-AIU/SPC, P1 Technologies) was implanted at the same coordinates as the electrodes implanted in the animals used for the optogenetic experiments. Seizures were observed from days 12–33 after KA injection.

## Long-term EEG recordings of KA-induced spontaneous seizures
Recordings for the animal injected with intrahippocampal kainic acid started 12 days after the surgery to allow for epileptogenesis to proceed with development of spontaneous and recurrent seizures[58]. Animals were housed with a tethered connection for long-term 24/7 video and EEG recordings. Other than the tether, the animals were able to move freely and provided with food and water ad libitum during the entire recording duration. The EEG recordings were performed using the same technical parameters as used to record animals that underwent optogenetic stimulations. Spontaneous seizures were detected by reviewing the EEG recordings. Seizure severities were scored the same as in optogenetically-induced seizures by visually reviewing the synchronous video recordings captured with the USB cameras (C920, Logitech).

## Pulsogram
To examine the dynamics of EEG response patterns relevant to ictogenesis, we engineered a time-vs-time plot or pulsogram (Fig. 3 and Fig. 4), as a 2D, color-coded representation of EEG data to supplement traditional time-frequency analysis. In the pulsogram for optogenetic recordings, the y-axis represents the EEG response time-locked to each stimulus pulse. The x-axis depicts the chronological progression of stimulation pulses (extrapolated at the pulse frequency beyond the

stimulation period). For spontaneous recurrent seizures, a time-locked stimulus was not available to construct the pulsogram. Therefore, a spike-detection algorithm was first used to detect EEG spikes greater than 6 median-absolute-deviations above zero. Inbuilt functions in MATLAB were used to detect peaks after smoothing. To keep only the first peak of a multi-phasic discharge, peaks that were less than 20 ms from the preceding peak were discarded. The pulsogram was then constructed using the detected spikes (Fig. 7b–d and Supplementary Fig. 11). Here, the y-axis represents the EEG response about the spike. The x-axis depicts the chronological progression of detected spikes. The pulsogram allowed simultaneous visualization of EEG response patterns on the pulse timescale (milliseconds) and the evolution of the EEG pattern on the epoch timescale (seconds). The pulsogram provided an innovative analysis of the evoked neuronal response.

EEG power was analyzed in the same format as the pulsogram (Fig. 5d, e). The instantaneous power was estimated from the Hilbert transform of the EEG. The instantaneous power was smoothed (10 ms, 2nd order Savitzky–Golay FIR). Then, the pulsogram was constructed as before but with the smoothed instantaneous power. To examine the power around the divergent point across seizures (Fig. 5f–h, Supplementary Fig. 5), the induction and reverberant phases were rescaled to equal lengths for all seizures for averaging.

## Pulsogram Response Trajectory Analysis

The pulsogram can be conceptualized as a response trajectory over time in an $n$-dimensional space, with each vertical strip (neuronal response to 1 pulse of stimulation) of the pulsogram represented as an $n$-dimensional vector. The trajectories were further analyzed using PCA, dynamic time warping and angular segmentation.

The pulsogram underwent a three-stage pre-processing prior to further trajectory analysis. Firstly, it was smoothed along the y-dimension (2 ms, 2nd order, Savitzky–Golay filter). The second stage of processing was designed to attenuate the contribution of spurious signals (motion artifacts) and the overwhelming paroxysmal activity. For this, each vertical strip was normalized by a scaling factor $\kappa[x]$ based on the response power of each $y$-strip, regulated by the average across all strips:

$$\kappa[x] = \sqrt{\frac{v[x] + v_0}{2}} \tag{1}$$

$$v[x] = \sum_{y \in [-5,50]ms} (s[y])^2 \tag{2}$$

$$v_0 = \frac{1}{n_{stim}} \sum_{x \in stim} v[x] \tag{3}$$

where $s[y]$ is the signal and $v[x]$ is the power of the EEG signal from −5 to 50 ms around each pulse. Regularization prevents excessive scaling of quiet activity and over attenuation to spurious activity. The last stage of preprocessing further smoothed the data x-dimension (2 s, 2nd order Savitzky–Golay filter). Subsequent analyses all employed the preprocessed pulsograms, analyzing on the y-segment from −5–50ms around each pulse.

## Principle Component Analysis (PCA) of trajectories

PCA was performed to visualize the pulsogram trajectory (Fig. 5b, c). For this visualization, PCA was performed on a pre-processed pulsogram of the entire recording. First and second principal components were extracted around each stimulation epoch for plotting.

PCA was also employed to compare the complexity of the trajectories (Fig. 6f and Supplementary Fig. 10). For this analysis, PCA decomposition was performed separately for each seizure using the 30 s stimulation segment of the processed pulsogram.

## Dynamic Time Warping (DTW) analysis

DTW was used to stretch and align two trajectories such that they are minimally separated in Euclidean distance[59,60]. The algorithm maintains the chronological order of the data points. Response trajectory (30 s stimulation segment of the processed pulsogram) between the response trajectories of the same recording were compared using DTW, performed using inbuilt function in MATLAB (dtw). The final separation distance between warped trajectories returned by the algorithm was averaged by the paired response type; similar trajectories are identified by smaller separation distances (Fig. 6c, d and Supplementary Fig. 8b, c). Variance in the separation distance measurements between recording sessions was minimized by normalizing distance by the average distance from Epoch 1 to all other 14 epochs.

The warped trajectories from the DTW were used to demonstrate the nesting of response phases between response types (Fig. 6a). The cumulative percentage of the trajectory warped against one another from the same recordings was calculated and averaged by the paired response type (Fig. 6b, c). Percentage values higher than the diagonal unity line indicates that the y-axis trajectory is used up faster than the x-axis trajectory in the warping process. This indicates that the y-trajectory has less advanced phases than the x-trajectory, i.e., the response in the x-trajectory encompasses the response in the y-trajectory and more.

## Angular segmentation analysis

Angular segmentation divided the response trajectory into segments by considering the turning angles between each $n$-dimensional data point (Supplementary Fig. 9). Turning angles are between 0° and 180° with sharper turns resulting in larger angles, representing larger changes in the pulse response. We used troughs to segment the response rather than peaks of turning angles; this is because troughs aligned better with visually identified segmentation points, and especially to the divergent and paroxysmal points. Specifically, the turning angles were calculated between the mean response over a window of 1 s before and after each point in time. Candidate segmentation points were initially identified with MATLAB's findpeaks function by finding troughs below a 90° turning angle with a minimum prominence angle of 30° (relative to neighbors). Heuristics that were used to eliminate minor troughs to arrive at the final segmentation are (i) troughs were separated by less than 0.67 s in time (Eq. 4), (ii) troughs deviated marginally in response (Eq. 5), and (iii) the trough was just a short excursion to a larger overall trend in the same trajectory direction, correlation coefficient greater than or equal to 0.9 (Eq. 6). Expressed mathematically:

$$t[p+1] - t[p] > 0.67 \tag{4}$$

$$\bar{\mathbf{R}}[p+1,p] - \bar{\mathbf{R}}[p,p-1] > \frac{d}{\sqrt{2}} \tag{5}$$

$$corr(\bar{\mathbf{R}}[p+1,p], \bar{\mathbf{R}}[p,p-1]) < 0.9 \tag{6}$$

where $t[p]$ is the time stamp of the candidate point $p$; $\bar{\mathbf{R}}[p,q]$ is the mean response between segments $p$ and $q$, calculated by Eq. 7 in vectorized form of the pulse response ($\mathbf{R}[x]$) at time $x$; and $d$ is the average Euclidean distance between data points (Eq. 8):

$$\bar{\mathbf{R}}[p,q] = \frac{1}{n_{ij}} \sum_{x=t[p]}^{t[q]} \mathbf{R}[x] \tag{7}$$

$$d = \frac{1}{n_x} \sum_x |\mathbf{R}[x+1] - \mathbf{R}[x]| \tag{8}$$

**Article**

Angular segmentation was performed on the response trajectory during the 30 s stimulation (extracted from the processed pulsogram). The number of segments estimated were averaged (in logarithm space) across responses of the same type (Fig. 6e and Supplementary Fig. 10). Angular segmentation results were cross-verified by segmenting the trajectory using the Ramer–Douglas–Peucker algorithm[61]. This algorithm finds a subset of data points that sufficiently represented the original trajectory such that all remaining data points are confined within a specified perpendicular distance from the simplified trajectory (Supplementary Fig. 10)[62,63].

### Reporting summary
Further information on research design is available in the Nature Portfolio Reporting Summary linked to this article.

## Data availability
The processed data to reproduce the statistics and figures used in this study are available in the Zenodo database under accession code https://doi.org/10.5281/zenodo.8274424[64]. Raw data are available from the corresponding author upon request due to data size. Source data are provided with this paper.

## Code availability
The code to reproduce the statistics and figures used in this study are available in the Zenodo database under accession code https://doi.org/10.5281/zenodo.8274424[64]. Full custom code base to review and process raw data is available from the corresponding author upon request.

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

## Acknowledgements
We thank Dr. David Barker for his expertise in equipment set up. We also thank Xiao Deng, Jaskaran Grewal, Alexandria Lo, Ayaan Memon, and Katherine Park who assisted with analysis of the data. Additionally, we thank the research staff from Louisiana State University: Omar Ashraf, Hayden Craig, Lana Larmeu, Benjamin Barker, Cade Stephenson, Derrick Murcia, and Brady Howard. Lastly, we thank Dr. Albert E. Glasscock for advising on our initial implementation of optogenetics.

## Author contributions
Conception and design of the study: H.S., S.C.C. Acquisition and ana-lysis of data: J.M., F.C.T., T.H., D.A.B., D.J.V., K.O., S.S., S.C.C. Drafting the manuscript and figures: J.M., S.C.C., H.S.

## Competing interests
The authors report no competing interests.

## Additional information

**Peer review information** *Nature Communications* thanks Maxime Lév-esque, David Klorig and the other, anonymous, reviewer for their con-tribution to the peer review of this work. A peer review file is available.

