## [Peer Review File · Nature Communications]

Ictogenesis Proceeds Through Discrete Phases in Hippocampal CA1 Seizures in MiceREVIEWER COMMENTS

Reviewer #1 (Remarks to the Author):

In this manuscript, Meuller et al investigate the progression of network reverberations during stimulus-induced seizure activity. To accomplish this, they optically stimulated pyramidal neurons in CA1, while simultaneously recording EEG. Such stimulation regularly induced seizures in a frequency-dependent manner. The primary findings were:

1. During each recording session, the first stimulus epoch of the day was more likely to elicit seizure than later stimuli.
2. Seizures induced later during a recording session were more severe.
3. In stimulus epochs that induced seizure, there was a stereotyped progression of activity recorded electrographically: first (induction phase) there was an immediate discharge only after each light pulse, then (reverberant phase) a secondary discharge emerged ~20ms after the immediate discharge, finally (paroxysmal phase) there was a high level of activity that was not phase-locked to the stimulation pulses
4. The paroxysmal and reverberant phases were always entered before seizure onset, but they were not sufficient to generate a seizure.

The study demonstrates a unique tool for analyzing the progression of reverberant activity during stimulus-induced epileptiform activity. The data contained in the manuscript are interesting, analyzed appropriately, and presented clearly. However, the correlation between optogenetic stimulus-evoked activity and spontaneous seizure generation remains unclear. The new approach to measuring network excitability appears to be a useful tool, but without a clear connection to spontaneous seizure dynamics, it may not have the impact required for a general audience.

Major Concerns:

1. Enthusiasm is reduced overall by the lack of clarity on what is being modeled with optogenetic stimulation of putative excitatory cells. The discussion suggests that the stimulation simulates the origination of the ictal core, however 10-20Hz synchronous stimulation of the recorded region simulates a pathological input to the region, perhaps mimicking an ictal core that emerged elsewhere. It could be argued that this is a model of seizure propagation, where the optogenetic stimulation simulates input from an approaching ictal core. In this case, it would be useful to analyze electrical recordings from a model of spontaneous seizure to look for a transition from single discharges to 20ms reverberant activity as the ictal core approaches the recording electrode.
2. It is not obvious what neural circuitry is activated during stimulation. It is reported that CA1 pyramidal neurons are activated by the optical stimulus, but the recording electrode is in the s.r. / slm region of the hippocampus. What is the presumed (polysynaptic?) pathway that CA1 activation produces synaptic activation in the distal dendrites of CA1?
3. The first two primary findings highlighted above are well-studied phenomenon. Given that there is only 90s of rest between bouts of stimulus, the "seizure-resistance" conferred by the first seizure of the day is consistent with post-ictal depression. Similarly, the worsening of seizures with successive stimulation bouts are consistent with short-interval kindling. It may be useful to more consistently refer

to these competing phenomena with established terminology. While these effects are certainly important to document, it also leaves the description of reverberant and paroxysmal responses to stimulation as the truly novel finding of this work.

Minor Points:

Authors discuss differences in “naïve” seizures versus later evoked seizures. However, these are not truly naïve seizures as the same animals are recorded from multiple times. In fact, figure 3 shows that “naïve” seizures are different on subsequent recording days. Alternate terminology may make this distinction clearer (perhaps first evoked seizure per day).

It would be interesting if the authors performed the same comparisons of trajectories and segmentation across the recordings obtained on different days to test how stable these patterns of activity are over time.

Legibility of figures could be improved throughout. Often too small or relevant labels in later panels of figure.

Figure 4 is very important but also confusing. The arrow from “A” to “C” can be removed. Labeling in “C” could be improved. It is not immediately clear that the Y-axis of time is per pulse. Consistent labeling of the time could help. (0 ms , 20 ms and 60 ms time points should be done in a uniform way). And deemphasize the labels for immediate versus secondary discharge phases. G and H should be labeled with latency and duration respectively as these are more universally understood terms. Add time label to top panel in 5A (with the example evolving seizure). Include the line as in figure 4 to show where the secondary discharge phase begins (i.e. at 20 ms). Colors in 5G are difficult to distinguish and poorly labeled. Are the 2 colors per type 10 vs 20 hz stimulation?

Trajectory analysis: Can authors clarify what is meant by attenuating the contribution of spurious signals during the preprocessing step? Are these some type of artifact?

Reviewer #2 (Remarks to the Author):

It is still unclear how neuronal activity evolves from an interictal to an ictal state. In their study, Mueller et al studied ictogenesis using optogenetics in mutant mice. They activated excitatory pyramidal cells in the CA1 region of the mouse hippocampus to induce hypersynchronous ictal discharges, that were further analysed using computational techniques. They identified three phases of ictogenesis, namely the induction phase, the reverberant phase and the paroxysmal phase. According to these authors, these steps would be necessary for local neuronal activity to become a seizure. Methodology is sound, experiments were carefully designed, and authors performed an in-depth analysis of data. These are innovative findings and the hypothesis of seizures progressing through different steps merits further investigation.

My main comment is that these findings can so far only be applied to the kindling model. It is unknown if such phenomenon is observed in spontaneous seizures in animal models of epilepsy (kainic acid, pilocarpine) or in epileptic patients. If it can be demonstrated that such steps exist in spontaneous seizures, it would be a major finding that will help us to better understand network mechanisms that underlie seizure generation.

It should be mentioned in the title of the manuscript that seizures were kindled with optogenetic stimulation.

It is unclear if naïve seizures were observed on the first day of stimulation or if multiple sessions of stimulation were needed to trigger ictal activity, since it is unlikely that seizures will be triggered in a non-epileptic animal. It usually takes a few days of stimulation to induce seizures with electrical or optogenetic stimulation in a naïve animal. Seizures can be rapidly triggered with optogenetic stimulation, but in animals that were pre-treated with chemoconvulsants and that are already epileptic.

Please specify why viral vectors with ChR2-eYFP under the CamKIIa promoter to target putative glutamatergic neurons were injected in PV-Cre mice. Wild-type mice could have been used, since there might be differences in susceptibility to seizures between genetic backgrounds.

It is unclear why bipolar electrodes were used since it is mentioned that (line 457) « the EEG response analyzed was taken between one of the CA1 bipolar electrodes and the distant reference to maintain consistency with monopolar implants ». Why bipolar and monopolar montages were used if bipolar recordings were used as monopolar recordings?

Figure 1 : A naïve seizure is shown, followed by two flat non-seizure responses, a short breakthrough and a long breakthrough seizure. Were these seizure types always occurring in the same order? If that was the case, I believe that these responses simply reflect the incapacity of neuronal networks to achieve levels of excitation and hypersynchrony necessary to generate ictal discharges during optogenetic stimulation. I agree with the authors that the first naïve seizure emerges from a brain that was unaffected by stimulation and that the following refractory period after the first seizure prevents further seizures from occurring. However, since the delay between stimulation epochs is short (90 s), it is unlikely that seizures will be triggered with every stimulation epoch. As shown in figure 1, the following two stimulation epochs triggered a flat response (no seizure), followed by a breakthrough short and a breakthrough long seizure. As stated by the authors, naïve seizures are characterized by complex EEG activity that progresses rapidly through different phases of ictogenesis, are less similar to non-seizure responses and recruits multiple neuronal ensembles. Naïve seizures therefore appear to be more similar to spontaneous seizures occurring in epileptic patients and animals compared to breakthrough seizures. The entire analysis could therefore be applied to naïve seizures only, in a protocol in which the delay between stimulation epochs is increased to more than 90s.

Figure 5 : It is unclear to me what the PCA brings to the analysis. For instance, although the induction phase of seizure types looks similar on the pulsograms (panel A), it appears really different with the PCA

analysis (panel B). Would it be possible to show an average PCA with multiple seizures?

It is mentioned in the discussion that « naïve and breakthrough seizures likely engage different cellular and network mechanisms to occur (table 2) ». Differences shown in table 2 (duration, Racine scores, incidence, etc.) cannot be used to argue that these two types of seizures involve different mechanisms of generation. Single-unit recordings would be necessary to establish such difference. Since they were both triggered with optogenetic stimulation of principal cells in CA1, I believe that they are generated by similar mechanisms.

Reviewer #3 (Remarks to the Author):

The authors present a novel paradigm for the study of evoked seizure activity including several helpful analysis techniques. Using optogenetic stimulation to induce seizures allows for analysis of the evoked population activity as the seizure progresses, giving some indication of the changing dynamics within the circuits involved. By repeatedly inducing seizures over time, the authors were also able to study the post-ictal state and its effect on seizure likelihood. Aided by a unique visualization technique to studying seizure progression, the pulsogram, the authors identified several discrete phases of ictogenesis for further characterization. Overall this is an interesting study with several new and useful approaches described. The pulsogram is very impressive and makes it quite easy to visually identify various features in the stimulus evoked response. The categorization of discrete “phases” of seizures and types is interesting and could form the basis for future studies to further characterize the underlying mechanisms responsible. The DWT and PCA trajectories are nice additions and help to further illustrate the differences between seizure phases. The increased complexity of naïve seizures vs. break-through is a fascinating finding that begs for deeper investigation. There are, however, a few methodological issues worth pointing out which may change the interpretation of some of the experiments and should be addressed or at least highlighted in the discussion (comments below).

Please clarify how light levels were selected. Did light power vary between animals or experiments? The authors provide a range for light intensity on lines 419-426 but it's not clear why light levels varied over such a large range (0.77-2.49 mW). If the variation is systematic (different intensities for each animal or experiment), please describe. If it is uncontrolled fluctuation, the range seems a bit large, especially for an LED source. Please provide some measure of variation (ie. SD, 95% CIs) for the average light power used. Was there any attempt to measure seizure thresholds in terms of light intensity prior to choosing the intensity used for each animal?

Viral expression of ChR2 builds over time before leveling out at ~40 days post injection (1,2), depending on promoter and target cells. The authors begin experiments after 14 days meaning that the expression levels of the opsin were lower for experiments run earlier compared to later. This could be confounding for experiments that compare results over time, or were conducted in the same animals on different days (Fig 1. D,E, Fig. 3 C,D). In the absence of a dedicated experiment tracking the opto-response over

days and weeks for the preparation used, the authors might be able to detect this effect as a change in the amplitude or slope of the short latency opto-response for the first pulse of the first stimulus on each recording day. I would be particularly concerned if the different seizure frequencies were tried progressively from 1Hz to 20Hz over the course of several weeks. If they were randomized or repeated with identical results, I would be less concerned. I did notice the 20Hz sessions are circled in Fig. 3C,D but it's not clear why the various frequencies were delivered when they were. Please specify and include labels (perhaps on a separate plot) for the other two frequencies as well (5 and 10Hz). Also, please include a description of how the order of the experiments was chosen in the results or methods and the caveat above about Chr2 expression in the discussion.

How many days were the recording sessions spread over? It might be nice to see a plot like 3C,D with days after the first stimulation on the x-axis. The authors state that the focus here is not on kindling despite using a kindling-like design but it would help with interpretation of the data presented to know if there were any multi-day breaks between recording sessions. That might explain some of the fluctuations in the Racine stage. Unfortunately, it also might serve as a contaminating variable in the pooled data comparing the results of various stimulation frequencies. More clarity on these points is requested.

Males and females were used, were there any sex differences?

Other Suggestions:

Supplementary Fig 3 is a helpful depiction of the pulsogram, please consider merging this figure with figure 4 in the main text, or creating a new figure specifically depicting the process.

It might be nice to have example raw traces from each phase to help readers get a better sense of how they translate to the pulsogram. Perhaps rather than the train in fig. 4a, which is a bit too zoomed-out to see what's going on, a single zoomed-in interpulse trace from each phase could be displayed with a similar indication of where in the trace below it occurred.

Please include an indication of time on the x-axis for fig. 2D & E (along with labels there already).

When using the term "phase" please specify you are referring to seizure/response phase (instead of phase as in the fraction of a cycle) in the Fig. 6A legend.

The use of past tense seems unnecessary in the sentence about DTW on lines 241-243.

Line 811, did you mean same stimulation or same recording session (multiple stimulations)?

Fig. 3 could be re-arranged to use space more efficiently.

While the use of PV-Cre mice is presumably related to experiments outside of the scope of the current work, the authors should be aware that Cre expression can change the phenotype of transgenic mice

(e.g. the increased seizure sensitivity of VGAT-Cre mice (3)). It's possible the results in a background strain could vary from what is presented here.

The electrode and fiber are larger than necessary for these experiments. In order to minimize damage to the brain and its effects on seizure activity, it will be important to minimize the size of the implanted hardware for future studies.

The sampling rate (3kHz) and BP filter (3-1000 Hz) seem low given the detailed analysis of optogenetic pulse responses performed. Many features of seizure / spike activity are present at higher frequencies and a lowpass of 1kHz would be expected to create significant distortion of the spike shape. The authors might try opening it up a bit to see if the extra detail adds to the texture of the pulseogram. Ideally these types of experiments would be acquired at a higher sampling rate, 5 or 10 kHz.

References

1. Wang, J. et al. Integrated device for combined optical neuromodulation and electrical recording for chronic in vivo applications. *J. Neural Eng.* 9, 016001 (2012).
2. Jayaprakash, N. et al. Optogenetic Interrogation of Functional Synapse Formation by Corticospinal Tract Axons in the Injured Spinal Cord. *J. Neurosci.* 36, 5877–5890 (2016).
3. Straub, J. et al. Characterization of kindled VGAT-Cre mice as a new animal model of temporal lobe epilepsy. *Epilepsia* 61, 2277–2288 (2020).

COMMENTS

Reviewer #1 (Remarks to the Author):

In this manuscript, Meuller et al investigate the progression of network reverberations during stimulus-induced seizure activity. To accomplish this, they optically stimulated pyramidal neurons in CA1, while simultaneously recording EEG. Such stimulation regularly induced seizures in a frequency-dependent manner. The primary findings were:

1. During each recording session, the first stimulus epoch of the day was more likely to elicit seizure than later stimuli.
2. Seizures induced later during a recording session were more severe.
3. In stimulus epochs that induced seizure, there was a stereotyped progression of activity recorded electrographically: first (induction phase) there was an immediate discharge only after each light pulse, then (reverberant phase) a secondary discharge emerged ~20ms after the immediate discharge, finally (paroxysmal phase) there was a high level of activity that was not phase-locked to the stimulation pulses
4. The paroxysmal and reverberant phases were always entered before seizure onset, but they were not sufficient to generate a seizure.

The study demonstrates a unique tool for analyzing the progression of reverberant activity during stimulus-induced epileptiform activity. The data contained in the manuscript are interesting, analyzed appropriately, and presented clearly. However, the correlation between optogenetic stimulus-evoked activity and spontaneous seizure generation remains unclear. The new approach to measuring network excitability appears to be a useful tool, but without a clear connection to spontaneous seizure dynamics, it may not have the impact required for a general audience.

Major Concerns:

1. Enthusiasm is reduced overall by the lack of clarity on what is being modeled with optogenetic stimulation of putative excitatory cells. The discussion suggests that the stimulation simulates the origination of the ictal core, however 10-20Hz synchronous stimulation of the recorded region simulates a pathological input to the region, perhaps mimicking an ictal core that emerged elsewhere. It could be argued that this is a model of seizure propagation, where the optogenetic stimulation simulates input from an approaching ictal core. In this case, it would be useful to analyze electrical recordings from a model of spontaneous seizure to look for a transition from single discharges to 20ms reverberant activity as the ictal core approaches the recording electrode.

Response: We thank the reviewer for providing an extensive review of our manuscript and an alternative perspective to our stimulation model. We believe we are modelling a focal hypersynchronous source with our optogenetic stimulation and capturing how a seizure developed from this focal hypersynchronous source, i.e., ictogenesis. It is conceivable that our model may simulate pathological input into CA1 instead of CA1 neurons spontaneously generating 10-20Hz activity as the seizure onset zone. As we discussed in the eighth paragraph of the discussion, and as the reviewer pointed out, the development of the ictal core shares similarities to the recruitment of additional neural tissue into the ictal core (seizure propagation). The challenges in differentiating these two scenarios are partly due to the limited spatial resolution in our recording device. Whether what we are modeling with optogenetic stimulation is the start of the ictal core or propagation from an ictal core cannot be determined with the current data set. What we are trying to convey with this study is that, leading to a seizure onset, the LFP recording in a certain area of the brain undergoes a stereotypical and stepwise change that can be delineated with the computational approaches

proposed by us. We have added the statement that what we are observing could be a model of seizure propagation to the eighth paragraph of the discussion to reflect the reviewer's comment.

Furthermore, we agreed with the reviewer's suggestion of carrying out our analysis with spontaneous seizures. Hence, we have added a subsection containing new results and a new figure adopting our analysis to spontaneous seizures. We have adopted the approach of the pulsogram in a murine model of temporal lobe epilepsy resulting from kainic acid-induced epileptogenesis and found a similar phenomenon in the LFP recordings from these animals. We believe that this provides further evidence of stepwise ictogenesis in spontaneous seizures as was detected in optogenetically-induced seizures. We have added the analysis to Fig. 7, Supplementary Fig. 11, the last subsection of the results, and discuss it in the first, fifth and last paragraphs of the discussion.

2. It is not obvious what neural circuitry is activated during stimulation. It is reported that CA1 pyramidal neurons are activated by the optical stimulus, but the recording electrode is in the s.r. / slm region of the hippocampus. What is the presumed (polysynaptic?) pathway that CA1 activation produces synaptic activation in the distal dendrites of CA1?

Response: Thank you for your comment. Initially, we thought we were primarily stimulating and recording from the same CA1 neurons that were excited by our optrode apparatus. As described in the "Viral injection and optrode implantation surgery" subsection of the methods, the design of the optrode was such that the electrode was placed where we expected to measure the electrophysiological response of the neurons directly under optogenetic stimulation.

As already stated in the fifth and sixth paragraphs of the discussion, we speculate, based on the delay of the secondary discharge, that the optogenetic activation could have crossed-over via the commissure to activate the contralateral EC-CA1 pathways before looping back to the ipsilateral EC-CA1 pathways. The brain state was gradually modified by the continual pulse train along this pathway leading to the appearance of the secondary discharge.

In response to your comment, we reconsidered the immediate discharge and have added the fourth paragraph of the discussion. Here, we discuss that we believe the response (immediate discharge) we measured with our electrode design upon individual optogenetic stimulations likely contained activity from a polysynaptic pathway.

3. The first two primary findings highlighted above are well-studied phenomenon. Given that there is only 90s of rest between bouts of stimulus, the "seizure-resistance" conferred by the first seizure of the day is consistent with post-ictal depression. Similarly, the worsening of seizures with successive stimulation bouts are consistent with short-interval kindling. It may be useful to more consistently refer to these competing phenomena with established terminology. While these effects are certainly important to document, it also leaves the description of reverberant and paroxysmal responses to stimulation as the truly novel finding of this work.

Response: Thank you for confirming our suspicions. In response to your comments, we have modified the results subsections where we present data regarding seizure-resistance and the rapid increases in kindling so that they use the established terminology as suggested.

Additionally, results subsection titles where these terms are introduced have been changed to “Nascent vs Breakthrough Seizures: Stimulation can overcome post-ictal depression to induce seizures” and “Post-ictal depression and resultant seizure resistant state paradoxically associated with more severe breakthrough seizures.” We have removed the section header “Higher seizure resistant state was accompanied by faster kindling”. Likewise, we believe that presentation of these kindling effects is necessary to understand the full scope of our results. We have also switched the kindling figure with the construction of a pulsogram figure since the pulsogram is central to detecting the reverberant and paroxysmal phases.

Minor Points:

Authors discuss differences in “naïve” seizures versus later evoked seizures. However, these are not truly naïve seizures as the same animals are recorded from multiple times. In fact, figure 3 shows that “naïve” seizures are different on subsequent recording days. Alternate terminology may make this distinction clearer (perhaps first evoked seizure per day).

Response: Thank you for your input. We agree that “naïve” may not properly convey the phenomenon we observed. We now use the word “nascent” to denote seizures occurring during the first epoch of a recording. We prefer not to use the term “first evoked seizure per day”, since the first evoked seizure always occurred during the first stimulation in all our recording sessions where stimulations induced seizures. When a seizure did not occur in the first epoch, no seizures occurred in the remaining epochs of a recording.

It would be interesting if the authors performed the same comparisons of trajectories and segmentation across the recordings obtained on different days to test how stable these patterns of activity are over time.

Response: Visual observation of our pulsograms indicated a clear evolution of the baseline response across weeks. This could be due to factors including fluctuations in the electrode impedance, continual maturity of the viral expression, and differences introduced by the disconnection and connection of the optic fiber over the time course of the experiments, etc. In general, the response was largely stable over several days. We have added a statement regarding this to the 2nd paragraph of the subsection of the results titled “Induction and reverberant phases are necessary but insufficient for ictogenesis.” To demonstrate the overall stability, we have included a series of seizure pulsograms longitudinally in a new Supplementary Fig. 7.

Legibility of figures could be improved throughout. Often too small or relevant labels in later panels of figure.

Response: Font size for all figures have been increased by 2pts.

Figure 4 is very important but also confusing. The arrow from “A” to “C” can be removed. Labeling in “C” could be improved. It is not immediately clear that the Y-axis of time is per pulse. Consistent labeling of the time could help. (0 ms, 20 ms and 60 ms time points should be done in a uniform way). And deemphasize the labels for immediate versus secondary discharge phases. G and H should be labeled with latency and duration respectively as these are more universally understood terms.

Response: Figure 4 has been revised as suggested.

Add time label to top panel in 5A (with the example evolving seizure). Include the line as in figure 4 to show where the secondary discharge phase begins (i.e. at 20 ms). Colors in 5G are difficult to distinguish and poorly labeled. Are the 2 colors per type 10 vs 20 hz stimulation?

Response: Figure 5 has been revised as suggested. Colors are intended to represent different time segments and seizure types and have been explicitly labeled as such in the revised figure.

Trajectory analysis: Can authors clarify what is meant by attenuating the contribution of spurious signals during the preprocessing step? Are these some type of artifact?

Response: Yes, these are motion artefacts from the animal. They are contained within a single vertical strip of the pulsogram, but do not affect the data analyzed in the prior or the following strip. In PCA analysis, they show up as spurious deviations from the main trajectory and detract from the intent of the analysis. Therefore, smoothing was performed to reduce their influence on the dynamic time warping algorithm. We have clarified that these are “motion artifacts” within the section titled “Pulsogram Response Trajectory Analysis.”

Reviewer #2 (Remarks to the Author):

It is still unclear how neuronal activity evolves from an interictal to an ictal state. In their study, Mueller et al studied ictogenesis using optogenetics in mutant mice. They activated excitatory pyramidal cells in the CA1 region of the mouse hippocampus to induce hypersynchronous ictal discharges, that were further analysed using computational techniques. They identified three phases of ictogenesis, namely the induction phase, the reverberant phase and the paroxysmal phase. According to these authors, these steps would be necessary for local neuronal activity to become a seizure. Methodology is sound, experiments were carefully designed, and authors performed an in-depth analysis of data. These are innovative findings and the hypothesis of seizures progressing through different steps merits further investigation.

My main comment is that these findings can so far only be applied to the kindling model. It is unknown if such phenomenon is observed in spontaneous seizures in animal models of epilepsy (kainic acid, pilocarpine) or in epileptic patients. If it can be demonstrated that such steps exist in spontaneous seizures, it would be a major finding that will help us to better understand network mechanisms that underlie seizure generation.

Response: Thank you for your comments and in-depth review of our manuscript. We have constructed the pulsogram in a murine model of temporal lobe epilepsy with spontaneous seizures resulting from KA-induced epileptogenesis and found a similar phenomenon of reverberant activity. We believe that this provides further evidence of a stepwise ictogenesis in spontaneous seizures similar to the phenomenon detected in optogenetically-induced seizures. We have added the analysis to the main body of the manuscript in the last subsection of the results and depicted the pulsogram constructed from spontaneous seizures in Fig. 7 and Supplementary Fig. 11. We also discuss these results in the first, fifth and last paragraphs of the discussion.

It should be mentioned in the title of the manuscript that seizures were kindled with optogenetic stimulation.

Response: Thank you for your feedback on the title. While a kindling effect was apparent and accounted for in the analysis, the goals of the paper were not to only investigate the effects of kindling. Since we added an analysis of spontaneous and recurrent seizures, we left the title unedited.

It is unclear if naïve seizures were observed on the first day of stimulation or if multiple sessions of stimulation were needed to trigger ictal activity, since it is unlikely that seizures will be triggered in a non-epileptic animal. It usually takes a few days of stimulation to induce seizures with electrical or optogenetic stimulation in a naive animal. Seizures can be rapidly triggered with optogenetic stimulation, but in animals that were pre-treated with chemoconvulsants and that are already epileptic.

Response: All data in the manuscript always contained the first recording of every animal included in this study. The light power and stimulation paradigm we employed for this study reliably induced a seizure during the first epoch of almost every recording session among all animals. The naïve seizures from the first stimulation session had Racine Score 0

(electrographic) or 1 (see Supplementary Fig. 4). We have clarified this in the first subsection of the results and added an experimental timeline (Supplementary Fig. 2).

Furthermore, the stimulation paradigm consistently elicited seizures in the first epoch in subsequent experiments (results not shown). We understand this presents a different finding to the reviewer's experience and expectations; we suspect this might be due to the fact that our light stimulation activated a sufficient number of neurons to induce seizures upon the delivery of the first bout of stimulation.

Please note that naïve seizures have been renamed to "nascent" seizures in the revised manuscript in response to comments from another reviewer.

Please specify why viral vectors with ChR2-eYFP under the CamKIIa promoter to target putative glutamatergic neurons were injected in PV-Cre mice. Wild-type mice could have been used, since there might be differences in susceptibility to seizures between genetic backgrounds.

Response: It is conceivable that there are differences in seizure susceptibility between PV-Cre and wild-type mice. We suspect that changes in seizure susceptibility may result in a change in overall seizure frequency in response to light stimulation. We believe these changes are unlikely to affect the mechanisms behind the ictogenesis we observed. We used PV-Cre mice in preparation for follow-up experiments to investigate the role of PV cells in ictogenesis. The results from this study served as foundational data for the follow-up experiments. We have added a note of this to the last paragraph of the first subsection of the methods as well as a citation where different susceptibility between PV-Cre mice and CaMKIIa-Cre mice to seizures was noted in a KA model.

It is unclear why bipolar electrodes were used since it is mentioned that (line 457) « the EEG response analyzed was taken between one of the CA1 bipolar electrodes and the distant reference to maintain consistency with monopolar implants ». Why bipolar and monopolar montages were used if bipolar recordings were used as monopolar recordings?

Response: Bipolar electrodes were used to compare LFP recordings from these electrodes with those from monopolar electrodes. Our preliminary analyses found no difference in the response we were interested in analyzing among these two types of recordings. Even with bipolar electrodes implanted, we still obtained LFP recordings similar to a monopolar recording electrode. To maintain consistency in data analysis across animals, this manuscript only included the monopolar data captured from the bipolar electrodes. We have added a note of this to the fourth paragraph of the subsection of the methods titled "Optogenetic stimulations and in vivo EEG recordings."

Figure 1: A naïve seizure is shown, followed by two flat non-seizure responses, a short breakthrough and a long breakthrough seizure. Were these seizure types always occurring in the same order?

If that was the case, I believe that these responses simply reflect the incapacity of neuronal networks to achieve levels of excitation and hypersynchrony necessary to generate ictal discharges during optogenetic stimulation. I agree with the authors that the first naïve seizure emerges from a

brain that was unaffected by stimulation and that the following refractory period after the first seizure prevents further seizures from occurring. However, since the delay between stimulation epochs is short (90 s), it is unlikely that seizures will be triggered with every stimulation epoch. As shown in figure 1, the following two stimulation epochs triggered a flat response (no seizure), followed by a breakthrough short and a breakthrough long seizure. As stated by the authors, naïve seizures are characterized by complex EEG activity that progresses rapidly through different phases of ictogenesis, are less similar to non-seizure responses and recruits multiple neuronal ensembles. Naïve seizures therefore appear to be more similar to spontaneous seizures occurring in epileptic patients and animals compared to breakthrough seizures. The entire analysis could therefore be applied to naïve seizures only, in a protocol in which the delay between stimulation epochs is increased to more than 90s.

Response: We appreciate the reviewer's keen observations and have contemplated similar questions when interpreting our results. We frequently observed a flat response on epoch 2 (but not always, see overall occurrence in Fig. 1F and G). Our experimental paradigm with the short inter-epoch time was by design to characterize the responsiveness of the underlying neural tissue during the post-ictal depression period, and how the same tissue subsequently regained seizure susceptibility.

We believe that the difference between naïve seizures and breakthrough seizures was an interesting finding that required further investigation. We therefore further characterized the different seizure types and reported these results (Supplementary Table 2). Determining whether naïve or breakthrough seizures were more similar to spontaneous seizures occurring among epileptic animals and patients was outside of the scope of this manuscript. However, we do plan on investigating this further: either computationally or with other experiments.

We did conduct additional experiments to investigate whether longer rest periods between bouts of stimulations may engender only naïve seizures. We did not include this data in this manuscript. Increasing inter-stimulation time would not necessarily yield repeated naïve seizures. The data from these experiments suggested that increasing the time did mitigate some effects of post-ictal depression, but modification to the ictogenic response pattern was still apparent up to 60 mins between stimulations.

Please note that naïve seizures have been renamed to "nascent" seizures in the revised manuscript in response to comments from another reviewer.

Figure 5: It is unclear to me what the PCA brings to the analysis. For instance, although the induction phase of seizure types looks similar on the pulsograms (panel A), it appears really different with the PCA analysis (panel B). Would it be possible to show an average PCA with multiple seizures?

Response: The PCA analysis is used to demonstrate that the different phases of ictogenesis have two main principal components (PC1 and PC2). When plotted, the phases group together in similar areas. The complexity of these principal components does differ among these responses with flat responses being the least complex and naïve seizure being the most complex. We have added panel C to Fig. 5, and Supplementary Fig. 6 showing both multiple PCAs over time and averaged PCAs. An explanation of these analyses was added to the second

paragraph of the subsection in the results titled “Induction and reverberant phases are necessary but insufficient for ictogenesis.”

Please note that naïve seizures have been renamed to “nascent” seizures in the revised manuscript in response to comments from another reviewer.

It is mentioned in the discussion that « naïve and breakthrough seizures likely engage different cellular and network mechanisms to occur (table 2) ». Differences shown in table 2 (duration, Racine scores, incidence, etc.) cannot be used to argue that these two types of seizures involve different mechanisms of generation. Single-unit recordings would be necessary to establish such difference. Since they were both triggered with optogenetic stimulation of principal cells in CA1, I believe that they are generated by similar mechanisms.

Response: Thank you for the feedback. We have updated the second paragraph of the discussion to reflect that single-unit recordings would be required to evaluate if different cellular and network mechanisms underlying the two different classifications of seizures.

Please note that naïve seizures have been renamed to “nascent” seizures in the revised manuscript in response to comments from another reviewer.

Reviewer #3 (Remarks to the Author):

The authors present a novel paradigm for the study of evoked seizure activity including several helpful analysis techniques. Using optogenetic stimulation to induce seizures allows for analysis of the evoked population activity as the seizure progresses, giving some indication of the changing dynamics within the circuits involved. By repeatedly inducing seizures over time, the authors were also able to study the post-ictal state and its effect on seizure likelihood. Aided by a unique visualization technique to studying seizure progression, the pulsogram, the authors identified several discrete phases of ictogenesis for further characterization. Overall this is an interesting study with several new and useful approaches described. The pulsogram is very impressive and makes it quite easy to visually identify various features in the stimulus evoked response. The categorization of discrete “phases” of seizures and types is interesting and could form the basis for future studies to further characterize the underlying mechanisms responsible. The DWT and PCA trajectories are nice additions and help to further illustrate the differences between seizure phases. The increased complexity of naïve seizures vs. break-through is a fascinating finding that begs for deeper investigation. There are, however, a few methodological issues worth pointing out which may change the interpretation of some of the experiments and should be addressed or at least highlighted in the discussion (comments below).

Please clarify how light levels were selected. Did light power vary between animals or experiments? The authors provide a range for light intensity on lines 419-426 but it's not clear why light levels varied over such a large range (0.77-2.49 mW). If the variation is systematic (different intensities for each animal or experiment), please describe. If it is uncontrolled fluctuation, the range seems a bit large, especially for an LED source. Please provide some measure of variation (ie. SD, 95% CIs) for the average light power used. Was there any attempt to measure seizure thresholds in terms of light intensity prior to choosing the intensity used for each animal?

Response: Thank you for your comments and the extensive review of our manuscript. We have added light power recording data split by mouse to Supplementary Table 1. We also included details on the light level fluctuation as a new final paragraph of the subsection in methods titled “Viral injection and optrode implantation surgery.” We have added Supplementary Fig. 11 to demonstrate that the least amount of light power necessary to elicit seizures was not consistent across mice. As a result of observations made during experiments, we selected a suprathreshold level of 2.0mW.

Viral expression of ChR2 builds over time before leveling out at ~40 days post injection (1,2), depending on promoter and target cells. The authors begin experiments after 14 days meaning that the expression levels of the opsin were lower for experiments run earlier compared to later. This could be confounding for experiments that compare results over time, or were conducted in the same animals on different days (Fig 1. D,E, Fig. 3 C,D). In the absence of a dedicated experiment tracking the opto-response over days and weeks for the preparation used, the authors might be able to detect this effect as a change in the amplitude or slope of the short latency opto-response for the first pulse of the first stimulus on each recording day.

Response: Thank you for this information. Indeed, the evolving viral expression could be a confound to the longitudinal effects that we observed. We plan to incorporate this consideration into our future experiment design. We have reviewed the dates of our experiments and found that all but 1 recording was made before 27 days post-injection. We have adjusted the sentence describing the amount of time from injection of AAV to the first

recording session in the first paragraph of the subsection of the methods titled “Optogenetic stimulations and *in vivo* EEG recordings”. We have also added the number of days between the viral injection and the first recording for each mouse to Supplementary Table 1 and a recording timeline as the new Supplementary Fig. 2. We believe that the evolving expression level of the ChR2 opsin was likely less of a confounding factor since most of the recordings were carried out later than we first reported.

Nonetheless, we have reported this possible limitation where the kindling results are presented in the last paragraph of the results subsection titled “Post-ictal depression and resultant seizure resistant state paradoxically associated with more severe breakthrough seizures.” We also added Supplementary Fig. 12, which presents the first pulse response peak recorded by mouse so the reader can interpret the change in the first pulse response with recording session.

I would be particularly concerned if the different seizure frequencies were tried progressively from 1Hz to 20Hz over the course of several weeks. If they were randomized or repeated with identical results, I would be less concerned. I did noticed the 20Hz sessions are circled in Fig. 3C,D but it's not clear why the various frequencies were delivered when they were. Please specify and include labels (perhaps on a separate plot) for the other two frequencies as well (5 and 10Hz). Also, please include a description of how the order of the experiments was chosen in the results or methods and the caveat above about ChR2 expression in the discussion.

Response: We added Supplementary Fig. 2 to show the entire experimental timeline and clarification to the third paragraph of the subsection in the methods titled “Optogenetic stimulations and *in vivo* EEG recordings.” As shown in Figure 1, Supplementary Table 1, and the new Supplementary Fig. 2, 10Hz stimulation was the main protocol used in this study and contributed to the bulk of the data. Seizure frequencies did not increase incrementally over the duration of experiments. 5Hz and 20Hz recordings were performed on arbitrary dates intermixed between 10Hz recordings. We examined whether these variations had a measurable effect on the 10Hz recordings, and the effect was minimal (Supplementary Fig. 2). The results were initially not included due to limited space in the manuscript.

How many days were the recording sessions spread over? It might be nice to see a plot like 3C,D with days after the first stimulation on the x-axis. The authors state that the focus here is not on kindling despite using a kindling-like design but it would help with interpretation of the data presented to know if there were any multi-day breaks between recording sessions. That might explain some of the fluctuations in the Racine stage. Unfortunately, it also might serve as contaminating variable in the pooled data comparing the results of various stimulation frequencies. More clarity on these points is requested.

Response: We added Supplementary Fig. 2 to show the entire experimental timeline. Since kindling was not the primary focus of the study, our experimental design did not adhere to a strict experimental protocol necessary to precisely characterize the progression of kindling. We presented the seizure severity with respect to recording session to convey that a kindling effect is present. The focus of this manuscript was on the phases of ictogenesis.

Males and females were used, were there any sex differences?

Response: We examined sex differences post-hoc and were not able to find any conclusive differences, so we pooled the data for the main analysis. We compared the probability of naïve seizure, rate of seizure per recording session, rate of flat response, Racine score, divergent point latency, and paroxysmal point latency against Gender. Gender was not identified as a significant factor among these comparisons (Wilcoxon rank-sum). We added this to the second paragraph of the subsection in the methods titled “Animals.”

Please note that naïve seizures have been renamed to “nascent” seizures in the revised manuscript in response to comments from another reviewer.

Other Suggestions:

Supplementary Fig 3 is a helpful depiction of the pulsogram, please consider merging this figure with figure 4 in the main text, or creating a new figure specifically depicting the process.

It might be nice to have example raw traces from each phase to help readers get a better sense of how they translate to the pulsogram. Perhaps rather than the train in fig. 4a, which is a bit too zoomed-out to see what’s going on, a single zoomed-in interpulse trace from each phase could be displayed with a similar indication of where in the trace below it occurred.

Response: Upon further consideration and feedback from you and other reviewers, we have switched Supplementary Fig. 3 with Fig. 3, so that Fig. 3 now shows the construction of the pulsogram, and Supplementary Fig. 4 are the seizure severity results, which are less central to the manuscript. We have revised Fig. 4 so that it includes traces from each phase and have removed the train.

Please include an indication of time on the x-axis for fig. 2D & E (along with labels there already).

Response: We have revised Figure 2 as suggested.

When using the term “phase” please specify you are referring to seizure/response phase (instead of phase as in the fraction of a cycle) in the Fig. 6A legend.

Response: Thank you for pointing out this clarification. We have clarified the figure caption.

The use of past tense seems unnecessary in the sentence about DTW on lines 241-243.

Response: Thank you for the correction. We have updated that sentence in the 3rd paragraph of section “Induction and reverberant phases are necessary but insufficient for ictogenesis” to present tense.

Line 811, did you mean same stimulation or same recording session (multiple stimulations)?

Response: We meant to say same recording session. We only compared responses from the same recording session with one another to minimize potential effects of longitudinal variation from recording setup. This sentence in Fig. 6D has been updated for clarity.

Fig. 3 could be re-arranged to use space more efficiently.

Response: We originally left space so that the caption could be placed where the white space is. In response to your comments and other reviewers, we have decided to switch Fig. 3 (Seizure severity, correlation with duration, and longitudinal changes) into the supplementary figures and instead use Supplementary Fig. 3 (Construction of a pulsogram) within the main text as Fig. 3.

While the use of PV-Cre mice is presumably related to experiments outside of the scope of the current work, the authors should be aware that Cre expression can change the phenotype of transgenic mice (e.g. the increased seizure sensitivity of VGAT-Cre mice (3)). It's possible the results in a background strain could vary from what is presented here.

Response: Thank you for the feedback. You are correct in pointing out that the main reason for using PV-Cre mice was to allow for follow-up experiments. We have added a statement regarding this as well as a citation where different susceptibility between PV-Cre mice and CaMKIIa-Cre mice to seizures was noted in a KA model to the second paragraph of the subsection in the methods titled "Animals."

The electrode and fiber are larger than necessary for these experiments. In order to minimize damage to the brain and its effects on seizure activity, it will be important to minimize the size of the implanted hardware for future studies.

Response: Thank you for the feedback. We will make sure that smaller hardware implants are utilized in future studies.

The sampling rate (3kHz) and BP filter (3-1000 Hz) seem low given the detailed analysis of optogenetic pulse responses performed. Many features of seizure / spike activity are present at higher frequencies and a lowpass of 1kHz would be expected to create significant distortion of the spike shape. The authors might try opening it up a bit to see if the extra detail adds to the texture of the pulsogram. Ideally these types of experiments would be acquired at a higher sampling rate, 5 or 10 kHz.

Response: Thank you for the feedback. We will increase our sample frequency in future recording experiments as suggested.

References

1. Wang, J. et al. Integrated device for combined optical neuromodulation and electrical recording for chronic in vivo applications. *J. Neural Eng.* 9, 016001 (2012).
2. Jayaprakash, N. et al. Optogenetic Interrogation of Functional Synapse Formation by Corticospinal Tract Axons in the Injured Spinal Cord. *J. Neurosci.* 36, 5877–5890 (2016).

3. Straub, J. et al. Characterization of kindled VGAT-Cre mice as a new animal model of temporal lobe epilepsy. *Epilepsia* 61, 2277–2288 (2020).

REVIEWERS' COMMENTS

Reviewer #1 (Remarks to the Author):

The authors have addressed all concerns raised in the initial review. In particular, new data was analyzed using the intrahippocampal kainate model of spontaneous seizure. While this was only performed in a single mouse, it demonstrates that the observed pulsogram transitions are not specific to stimulation induced seizure. The only minor criticisms remaining are below. Congratulations to the authors on this exciting manuscript.

Minor points:

1. For the seizures recorded from the mouse receiving IHK what was the Racine score? Or were all recorded seizures electrographic?

Figure 7 and Supplemental Fig 11 refer to pulse time. It would be clearer to change to spike or event time (as there was no pulse in these seizures).

Reviewer #2 (Remarks to the Author):

Authors have addressed all my comments.

Reviewer #3 (Remarks to the Author):

The authors were responsive to reviewer comments and the revised manuscript includes several important improvements and additional data (including supporting spontaneous seizure data, albeit from a single animal).

A few minor comments on the revised manuscript:

I'm not sure that "nascent" is better than "naïve" in this case. I second reviewer #1's suggestion of "first seizure of the day" as it is clear and unambiguous. Given that the term is used throughout, it could be shortened to "first seizure" for subsequent mentions.

Thank you for including the caveat regarding the time course of AAV mediated ChR2 expression. Good to know that the majority of recordings were performed after 27 days. It probably isn't necessary to include in the results section however, a quick mention in the discussion should be sufficient to alert the reader to the issue. Thank you for providing the complete time course of the experiments as well. Although the various frequencies were clustered in time and therefore potentially susceptible to the influence of increasing expression levels, the order of presentation makes it unlikely to present an issue.

Thank you for clarifying the light intensities used. It is a bit surprising that you did not observe a correlation between power and the evoked response, but I suppose it could be due to the limited range

of intensities used, the time between presentations, and the relatively low sample number.

Either way, it doesn't seem to be an issue for the data presented here given that the majority of stimuli used were at the same suprathreshold intensity (2 mW). Thank you for taking the time to verify that was the case and provide additional data.

A few suggestions for future studies:

It should be fine to use different light intensities for different animals to account for variations in viral expression and the implant. One approach would be to incrementally increase the light intensity to determine a seizure threshold for each animal and then select a suprathreshold stimulation level for subsequent experiments, similar to how seizure thresholds are determined for electrical kindling studies. I would be careful about changing light levels over time though unless you can identify what is causing the degradation in the source fiber. A bad fiber can vary unpredictably when bent and should be replaced. A few tips: A longer patch cable with some slack and a low inertia counter balance can be used to prevent excess strain on the fiber during seizures. For a counter balance, braided fishing line, two pulleys, and washers as weights work well. Use masking tape or sticky swabs to remove debris from fiber tips prior to connecting fibers each day. Be aware that LEDs can vary with temperature and it is common for the intensity to drop off a bit after the LED has been on for a while. This shouldn't be an issue for the seizure stim but might be an issue when using the power meter. Please note that your power meter is listed in the manuscript as a PM20C, I'm guessing you meant the PM20A (for visible light)?

Any steps taken to increase the precision and robustness of your light delivery system are well worth the effort given that stable light delivery is a prerequisite for accurate seizure threshold measurements.

REVIEWERS' COMMENTS

Reviewer #1 (Remarks to the Author):

The authors have addressed all concerns raised in the initial review. In particular, new data was analyzed using the intrahippocampal kainate model of spontaneous seizure. While this was only performed in a single mouse, it demonstrates that the observed pulsogram transitions are not specific to stimulation induced seizure. The only minor criticisms remaining are below. Congratulations to the authors on this exciting manuscript.

Response: Thank you for your thorough review of our manuscript, for the suggestions you provided, and for your consideration of our manuscript.

Minor points:

1. For the seizures recorded from the mouse receiving IHK what was the Racine score? Or were all recorded seizures electrographic?

Response: Thank you for pointing this out. We have added the Racine scores for each seizure to Supplementary Fig. 11. Additionally, we have clarified the first paragraph of the subsection of the results titled "Reverberance phase in CA1 preceded spontaneous seizures".

Figure 7 and Supplemental Fig 11 refer to pulse time. It would be clearer to change to spike or event time (as there was no pulse in these seizures).

Response: Thank you for your suggestion. We have update Figure 7 and Supplemental Fig. 11.

Reviewer #2 (Remarks to the Author):

Authors have addressed all my comments.

Response: Thank you for completing a thorough review and for the previous suggestions for improving our manuscript.

Reviewer #3 (Remarks to the Author):

The authors were responsive to reviewer comments and the revised manuscript includes several important improvements and additional data (including supporting spontaneous seizure data, albeit from a single animal).

Response: Thank you for carefully reviewing our manuscript and your thoughtful suggestions for improvements.

A few minor comments on the revised manuscript:

I'm not sure that "nascent" is better than "naïve" in this case. I second reviewer #1's suggestion of "first seizure of the day" as it is clear and unambiguous. Given that the term is used throughout, it could be shortened to "first seizure" for subsequent mentions.

Response: Thank you for your comment. We were hesitant to change the verbiage to “first seizure of the day” since preliminary results from other experiments suggested that seizures return to a similar state as “first seizure” after a certain amount of time. Since you and reviewer #1 suggested that we use “first seizure of the day” instead of naïve/nascent, we have updated the first mention of this type of seizure to “first seizure of the day” and subsequent mentions to “first seizures”. We have also updated the acronym from SzN to SzF.

Thank you for including the caveat regarding the time course of AAV mediated ChR2 expression. Good to know that the majority of recordings were performed after 27 days. It probably isn't necessary to include in the results section however, a quick mention in the discussion should be sufficient to alert the reader to the issue. Thank you for providing the complete time course of the experiments as well. Although the various frequencies were clustered in time and therefore potentially susceptible to the influence of increasing expression levels, the order of presentation makes it unlikely to present an issue.

Response: Thank you for your comment. We have removed mention of AAV mediated ChR2 expression timelines from the last paragraph of the Results subsection “Post-ictal depression and resultant seizure resistant state paradoxically associated with more severe breakthrough seizures,” and instead, added a brief mention of the kindling timelines and AAV expression consideration to the second section of the Discussion. Thank you for reviewing the timeline of our experiment. We also believe that the AAV expression is unlikely to change the overall interpretation of the frequency-dependent results that we observed.

Thank you for clarifying the light intensities used. It is a bit surprising that you did not observe a correlation between power and the evoked response, but I suppose it could be due to the limited range of intensities used, the time between presentations, and the relatively low sample number.

Either way, it doesn't seem to be an issue for the data presented here given that the majority of stimuli used were at the same suprathreshold intensity (2 mW). Thank you for taking the time to verify that was the case and provide additional data.

Response: Thank you for your careful review of our experimental protocol and for the suggestions that you provided to improve understanding of our manuscript. We also believe that the lack of correlation could be due to the points that you raised. We have ensured that intensity is controlled in subsequent and current experiments.

A few suggestions for future studies:

It should be fine to use different light intensities for different animals to account for variations in viral expression and the implant. One approach would be to incrementally increase the light intensity to determine a seizure threshold for each animal and then select a suprathreshold stimulation level for subsequent experiments, similar to how seizure thresholds are determined for electrical kindling studies. I would be careful about changing light levels over time though unless you can identify what is causing the degradation in the source fiber. A bad fiber can vary unpredictably when bent and should be replaced. A few tips: A longer patch cable with some slack and a low inertia counter balance can be used to prevent excess strain on the fiber during seizures.

For a counter balance, braided fishing line, two pulleys, and washers as weights work well. Use masking tape or sticky swabs to remove debris from fiber tips prior to connecting fibers each day. Be aware that LEDs can vary with temperature and it is common for the intensity to drop off a bit after the LED has been on for a while. This shouldn't be an issue for the seizure stim but might be an issue when using the power meter. Please note that your power meter is listed in the manuscript as a PM20C, I'm guessing you meant the PM20A (for visible light)?

Any steps taken to increase the precision and robustness of your light delivery system are well worth the effort given that stable light delivery is a prerequisite for accurate seizure threshold measurements.

Response: Thank you for your suggestions for improving our experimental recording setup. We handle our patch fibers with extreme caution between experiments and now control for intensity at the tip that is connected to the implant. We agree that increasing the robustness of our system is crucial and will incorporate your suggestions into future experiments.